# A Holistic Framework for Evaluating Adaptation Approaches to Coastal Hazards and Sea Level Rise: A Case Study from Imperial Beach, California

David Revell [1,*], Phil King [2], Jeff Giliam [3], Juliano Calil [4], Sarah Jenkins [5], Chris Helmer [6], Jim Nakagawa [6], Alex Snyder [7], Joe Ellis [8] and Matt Jamieson [9]

1   Integral Consulting (Formerly Revell Coastal), Santa Cruz, CA 95060, USA
2   Economics Department, San Francisco State University, San Francisco, CA 94132, USA; pgking@sfsu.edu
3   Economics Department, George Mason University, Fairfax, VA 22030, USA; jeffrey.giliam@gmail.com
4   Middlebury Institute of International Studies, Monterey, CA 93940, USA; juliano.calil@gmail.com
5   Economics Department, University of the Pacific, Stockton, CA 95211, USA; sjnkns97@gmail.com
6   City of Imperial Beach, San Diego, CA 91932, USA; chelmer@imperialbeachca.gov (C.H.); jnakagawa@me.com (J.N.)
7   U.S. Geological Survey, Santa Cruz, CA 95060, USA; agsnyder@usgs.gov
8   Marathon Construction Corporation, Lakeside, CA 92040, USA; Joe@marathonsd.com
9   Integral Consulting, Santa Cruz, CA 95060, USA; mjamieson@integral-corp.com
*   Correspondence: drevell@integral-corp.com; Tel.: +1-503-577-4515

**Abstract:** Sea level rise increases community risks from erosion, wave flooding, and tides. Current management typically protects existing development and infrastructure with coastal armoring. These practices ignore long-term impacts to public trust coastal recreation and natural ecosystems. This adaptation framework models physical responses to the public beach and private upland for each adaptation strategy over time, linking physical changes in widths to damages, economic costs, and benefits from beach recreation and nature using low-lying Imperial Beach, California, as a case study. Available coastal hazard models identified community vulnerabilities, and local risk communication engagement prioritized five adaptation approaches—armoring, nourishment, living shorelines, groins, and managed retreat. This framework innovates using replacement cost as a proxy for ecosystem services normally not valued and examines a managed retreat policy approach using a public buyout and rent-back option. Specific methods and economic values used in the analysis need more research and innovation, but the framework provides a scalable methodology to guide coastal adaptation planning everywhere. Case study results suggest that coastal armoring provides the least public benefits over time. Living shoreline approaches show greater public benefits, while managed retreat, implemented sooner, provides the best long-term adaptation strategy to protect community identity and public trust resources.

**Keywords:** sea level rise adaptation; vulnerability assessment; adaptation planning; cost–benefit analysis; public trust; ecological and recreation valuation; risk communication; living shoreline; adaptation pathways; coastal resilience; coastal hazards; coastal management

## 1. Introduction

Historically, coastal management decisions addressing coastal hazards respond to emergency storm damages from erosion and flooding. These decisions have focused on protecting existing assets—private development, public transportation, and water-related public infrastructure using coastal armoring. These short-term adaptation armoring responses typically protect assets at the expense of the long-term health of public trust resources, such as beach recreation and coastal ecosystems. Valuing public trust ecosystem services other than recreation and storm buffering is difficult and research is still relatively new [1–4]. As a result, the historical pattern has ignored these public trust values in coastal

management decisions. Ignoring these important resources is by default valuing them at *zero*; however, everyone knows that beach recreation and ecosystem services have *some* value. This framework addresses this oversight, assigning a dollar value to recreation and a replacement cost for ecosystem services to integrate and compare these public trust recreation and ecosystem services with upland damages during adaptation planning.

Over the last several decades, government agencies, researchers, and local coastal communities have increasingly recognized the need to adapt to sea level rise and associated coastal hazard threats as sea levels rise and projected damages increase [2,5,6]. Communities, researchers, and coastal management agencies have long advocated for the introduction of adaptation planning into the long-term urban planning framework [7], taking a long-range approach to prevent this ad hoc, emergency response method of coastal management [5]. As there are several divergent forms of adaptation—generally divided into protect via grey (armoring) or green (nourishment and living shoreline projects), accommodation, or managed retreat (Managed retreat from the authors public engagement experience is often a politically polarizing term, but refers to a graceful, equitable planned moving of development and infrastructure in harm's way over time accomplished in a wide variety of ways.)—coastal communities need physical and economic information to best decide their long-term approach between potential adaptation strategies [5,6,8].

In California, vulnerability assessments and adaptation planning have become a fundamental part of many planning processes. Scientific projections and state policy guidance continue to evolve and provide a variety of coastal hazard modeling resources, requirements for sea level rise considerations, and prioritized approaches for adaptation [2]. The state guidelines lay out a series of steps to evaluate a range of sea level rise scenarios and various coastal hazards to identify vulnerabilities. Once these vulnerabilities are identified, adaptation planning steps should occur to inform policy changes that can drive funding into local priorities and community vision.

A key part of adaptation planning and establishing local priorities should be a cost–benefit analysis (CBA) of options. Most previous CBAs (e.g., [9–14]) of coastal adaptation focus on the non-market values of recreation and/or storm prevention. For example in a widely cited study, Liu et al. [15] estimates the ecological benefits of beaches in New Jersey, but their "gap analysis" (p. 1276) notes the absence of studies looking at other ecological services provided by beaches other than recreation, "disturbance prevention", and "cultural and spiritual" values. Often a discussion of the ecological services of beaches is limited to these items. Similarly, Reguero et al. estimates the value of nature based solutions on the coast, but their benefits are measured in terms of flood prevention [13] alone.

In addition, many analyses of sea level rise impacts are nationwide or regional. While nationwide studies help capture news headlines, local scale assessments identify potential damages and adaptation approaches that consider individual community visions. Most of these assessments focus primarily on adapting development and infrastructure. An expanded focus on public trust beach recreation and ecosystem services is needed for local and site-specific studies to best guide communities' long-term adaptation planning [8,11–14,16,17]. Some studies account for armoring alone, or alongside "greener" alternatives, but established methods for comprehensive comparison of specific community's full range of adaptation alternatives are largely lacking. Comparison of adaptation methods is critical to minimizing losses and expenditures to coastal communities. One specific adaptation measure may reduce the risk from one hazard to one sector but cause issues elsewhere or lead to unintended secondary consequences. For example, coastal armoring can cause a long-term loss of beach by the placement loss or footprint of the structure, and the long-term drowning or passive erosion of the beach [4,6,18,19]; or a beach nourishment may affect stormwater drainage or alter lagoon and beach habitat functioning causing unintended flooding or detrimental habitat conditions.

Each adaptation approach has different implications for both upland property or infrastructure and the public trust beach. One innovation of this framework is that physical changes to the upland and beach from each adaptation approach are modeled and changes

in both widths are assigned economic values and the net benefits (costs minus benefits) are used as a basis of comparison to support adaptation decision making.

This study endeavors to address this gap in the literature and coastal management practice with a comprehensive benefit–cost framework that considers not only losses to upland property and infrastructure (public and private) but also avoided costs, future adaptation project lifecycle costs, as well as secondary economic impacts associated with public trust recreation and ecosystem services over time. It builds on past work estimating loss in beach recreation (non-market value) due to sea level rise and coastal storms, and modeling ecological value [20,21]. These estimates of loss of beach ecosystem services innovate by applying a replacement cost to beaches, similar to work done on wetlands [1].

With this framework, the paper aims to inform difficult local questions of how a coastal community can best adapt over time—minimizing the physical and financial impacts to upland property and infrastructure while considering important public trust beach recreation and ecosystem services. This framework provides for a comparison in economic net benefit terms over time that can support decision making for coastal communities around the world.

This paper begins with an introduction to the study site, and a summary of the three pronged risk communication approach that engaged City of Imperial Beach (City) departments, local experts, interested public, and elected decision makers to identify community priorities and guide adaptation planning. A discussion of the framework methodology follows, explaining the vulnerability and adaptation phases of the analysis. The methods describe both the physical modeling of upland and beach width and the economic methods that integrates the recreation and ecosystem services changes into the physical changes. Results from the specific Imperial Beach case study are applied via the framework to five adaptation approaches commonly considered in coastal management decisions. The discussion focuses primarily on how to improve the framework application and investigation to support community level adaptation decisions. Readers who are particularly interested in the impacts to the Imperial Beach community or site-specific details of the various analyses are encouraged to review the final consulting report [22].

## 2. Methods

This study focused on identifying existing and future coastal hazard vulnerabilities as well as feasible adaptation strategies for Imperial Beach. During the study, an outreach strategy engaged City department leadership and surrounding land managers to verify results and provide guidance and feedback on preferred adaptation strategies. During the project, the City, USC Sea Grant, and the consulting team worked with Tijuana River National Estuarine Research Reserve staff on a three-pronged risk communication strategy with a steering committee, public outreach, and City council deliberations. The steering committee, consisting of City department heads and a dozen key regional stakeholders (see acknowledgements), participated in seven half-day workshops, and provided substantive input on assumptions, data, adaptation priorities, and feedback on methods and results. Three public evening workshops including the largest in City history with 337 participants, presented information to the entire group then broke into smaller exploratory stations where participants could self-identify areas of interest. Finally, three different City council presentations and listening sessions informed future City actions.

The subsequent case study consists of a two-phased analysis (Figure 1). The first phase was a vulnerability assessment using the best available scientific projections of future extents of coastal erosion, coastal wave flooding, and tidal inundation. These coastal hazard projections were analyzed spatially in ArcGIS with asset data from the City to evaluate the exposure and potential damages from each hazard without adaptation. The second phase included an application of the conceptual framework and subsequent CBA on prioritized adaptation alternatives to compare and contrast the net economic benefits of each adaptation alternative over time. The following sections explain each step in this study, as outlined in the diagram below.

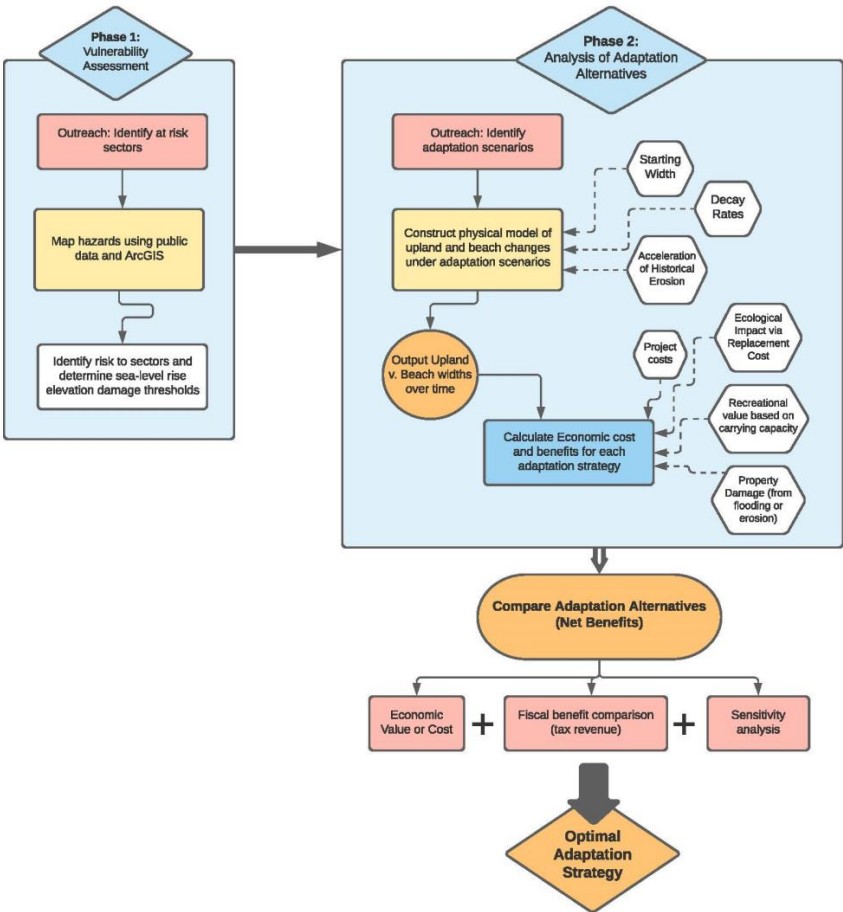

**Figure 1.** Adaptation framework process including Phase One (Vulnerability Assessment) and Phase Two (Analysis of Adaptation Alternatives).

### 2.1. Site Description

The City of Imperial Beach, the southwestern-most city in the continental United States, is located in San Diego County (County) and is surrounded by water on three sides, which exposes it to all types of coastal hazards as well as provides an opportune place to consider ecological services in adaptation planning (Figure 2). The City extends about 4 km along the coast. The developed land use is primarily residential, with approximately 27,000 residents [23], military, and a coastal ecotourism industry that relies on its coastline and proximity to adjacent habitats.

To the south is the Tijuana Estuary and River, natural wetland and dune habitats that are part of a National Estuarine Research Reserve that exposes the City to tidal inundation, lagoon, and fluvial hazards. To the north is San Diego Bay and a host of wetlands and old salt ponds that are slowly being restored that could reduce tidal inundation, which routinely occurs during extreme high "king" tides. To the west lies the City's prized beaches adjacent to the Pacific Ocean with its energetic wave climate that routinely causes coastal erosion and wave flooding. These waves and beaches provide a substantial recreation and economic draw for surfers, anglers, and an annual sandcastle contest and help to form the community identity.

These surrounding wetland and beach habitats support several threatened and endangered species including bird species, such as the clapper rail, least tern, western snowy plover, and least Bell's vireo, and plants, such as the salt marsh bird's beak and the grunion, a beach spawning fish. In addition, the various habitats provide home to countless other species, and perform many other ecological services.

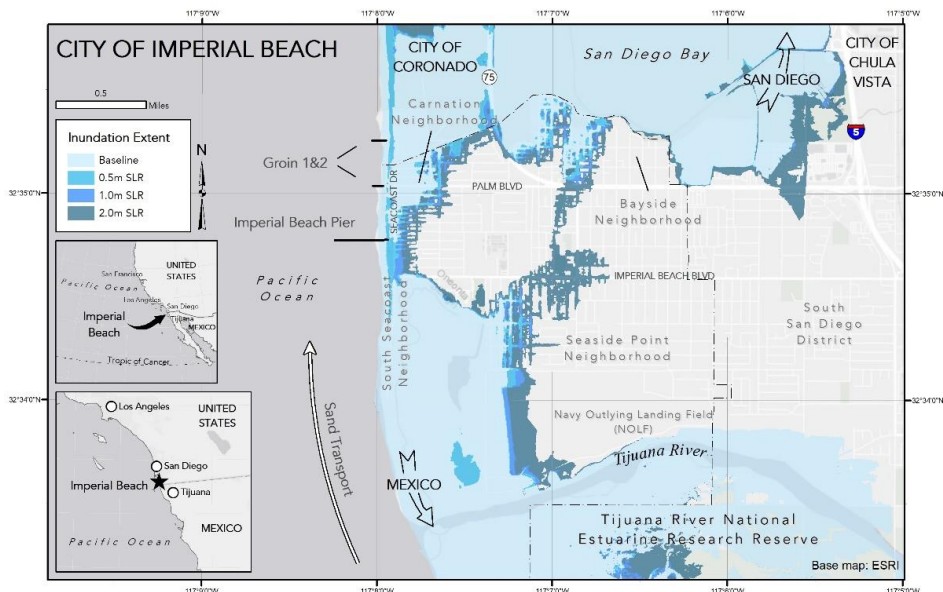

**Figure 2.** Case Study site of Imperial Beach in California, USA. Coastal hazard extents combined USGS CoSMoS 1.0, 3.0 Preliminary, and Department of Defense—SPAWAR.

The study site is located in the 61 km Silver Strand littoral cell that extends from Mexico to Point Loma north of the entrance of San Diego Bay. Tides in the area are characterized as mixed semi-diurnal tides with a spring tide range of about 1.6 m [24–27]. During the winter, wave energy from the northwest and west refracted around the Channel Islands move sand offshore and to the south [25,28]. During the summer time, distant south swells and local hurricane swells move sand to the north [26,27,29,30]. Projections of the 100-year recurrence interval for offshore significant wave heights are between 20 and 25 feet [24,27]. Sand transport is dominated by wave transport with seasonal reversals in transport direction occurring due to the direction of wave attack. While the littoral cell has a high gross sediment transport, the net transport of sand is from south to north in the littoral cell driven primarily by the strong exposure to summer south swells and hurricanes [25,26,29].

The City has about 1.4 miles of coastal armoring with 83 different structures, primarily revetments in various conditions [24]. These coastal armoring structures take up about 185,000 square feet of dry sand beach, or about four football fields. Two groins north of the pier are marginally effective at trapping sand moving from south to north. Since the 1940s, the City has a history of about a dozen beach nourishment projects providing more than 40 million cubic yards of sand to the beaches [27,29]. These nourishments have been critical to maintaining the public beaches in front of the coastal armoring (Figure 3).

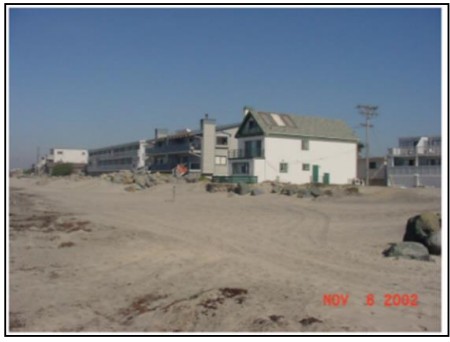
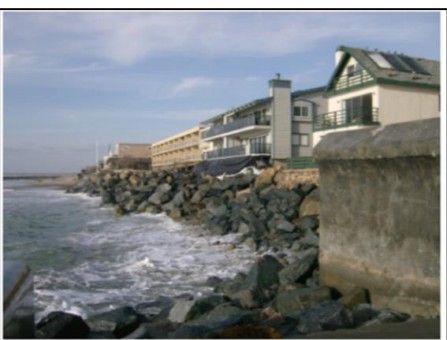

**Figure 3.** Impact of coastal armoring on public beach—with sand following nourishment (**left**), and without sand (**right**). Photo: J. Nakagawa.

*2.2. Phase One: Vulnerability Assessment*

The vulnerability assessment utilized available coastal hazard modeling projections for erosion and coastal wave flooding as well as asset information from the City. The geospatial analysis conducted in ArcGIS evaluated the exposure of coastal hazards on property and infrastructure in Imperial Beach using an intersect method.

One challenge of sea level rise and risk communication revolves around the uncertainty of when future various sea level rise elevations will be reached. For this study, based on available coastal hazard model projections, community input, and state guidance the study used a "2 m by 2100" as a worst-case sea level rise scenario for the vulnerability and adaptation analyses. This 2 m estimate is about 1 m higher than the IPCC [31] projection but relatively consistent with the California State guidance for medium risk aversion [2,32]. Sea level rise scenarios of 0.5, 1.0, 1.5, and 2.0 m were used to support City decision making over shorter time periods.

This study assessed three coastal hazards (Figure 4) based on site conditions and input from the community engagement process. These hazards were defined as:

- Coastal Flooding (a): Temporary flooding caused by a 1% annual chance storm wave event.
- Coastal Erosion (b): Permanent erosion resulting in loss of land from a 1% annual chance of wave erosion.
- Tidal Inundation (c): Periodic inundation caused during a high king tide event.

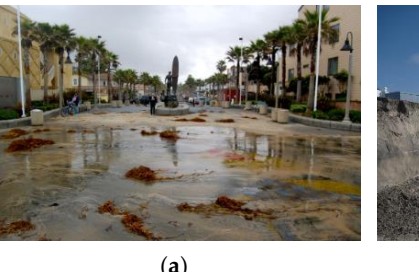 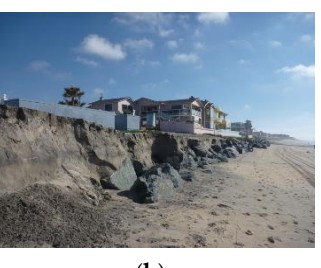 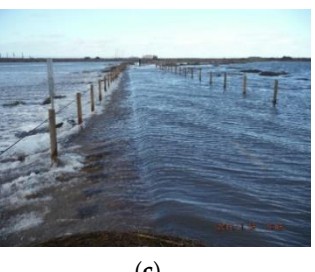

(**a**) (**b**) (**c**)

**Figure 4.** Coastal hazards (**a**) wave flooding, (**b**) coastal erosion, (**c**) tidal inundation.

This study relied on several sources of publicly available coastal hazard modeling projections that mapped extents of each hazard over a range of sea level rise scenarios. The maximum extents of each hazard were combined with the full extent of projected hazards shown in Figure 2, where one can see that with 2 m of sea level rise, the remnant geomorphic channels connecting the Tijuana River with San Diego Bay may become reactivated. Sources for the coastal hazard modeling data include:

U.S. Geological Survey (USGS) Coastal Storm Modeling System (CoSMoS): Merged coastal flood hazards from CoSMoS 1.0 associated with a historical ~10-year event from January of 2010 [33], with CoSMoS 3.0 (draft) projecting flood extents from a 100-year wave event at 0, 0.5, 1.0, and 2.0 m of sea level rise ([34]—used in submitted/draft form) (Draft CoSMoS results were used due to delays accessing final results during the project timeline).

- SPAWAR: Coastal erosion projections of 0, 0.5, 1.0, and 2.0 m [35].
- Asset Data: Infrastructure data from City of Imperial Beach, San Diego County, and the U.S. Environmental Protection Agency.

To capture the impact of these coastal hazards in Imperial Beach, the analysis focused on sectors based on City priorities identified through the steering committee engagement process. The vulnerability analysis focused on important sectors and relevant measures of impact including:

- Land Use—number of parcels and structures
- Roads—lengths
- Public Transportation—lengths and number of routes

- Wastewater—pipe lengths and pump stations
- Stormwater—drop inlets, pipe lengths, outfalls, and reduction in conveyance
- Schools—number of buildings and land area
- Hazardous Materials—number of businesses, underground tanks, cleanup sites.

Using spatial analysis in ArcGIS, exposure of various City identified priorities to each coastal hazard were mapped for different resource sectors over the various sea level rise scenarios. These maps were overlaid on the sectors identified by the City, and impact was determined based on the intersection of the sector assets with the three different coastal hazard types. Information on land use, stormwater, and schools was obtained from the City, roads public transportation and wastewater from San Diego County, and hazardous materials from the U.S. Environmental Protection Agency.

These sectors were chosen by the City and steering committee to reflect key areas of the concern for the community. For each sector, metrics useful to the City identified the exposure of the assets critical to each sector over time to each of the coastal hazards using geospatial analyses. In addition, potential economic damages and fiscal impacts were assessed from each hazard to compare with the different adaptation approaches.

Damage impacts to both private and public structures were estimated using standard U.S. Army Corps of Engineers (USACE) depth damage curves [10,12], which values damages as a percentage of the structures' replacement cost. Estimates of publicly owned structures and facilities were obtained from the City. Estimates of structure value obtained from the County level parcel data provided detailed geospatial data including size and exact location of each structure; most structures at risk are residential.

Beaches provide significant recreational value. The non-market value of Imperial Beach's beaches was estimated based on attendance estimates provided by the City (from lifeguard counts) multiplied by the "day-use" value of a trip to the beach. This day-use value technique is common for estimates of beach non-market value. Our benefits transfer method applied the Coastal Sediment Benefits Analysis Tool (CSBAT) developed in conjunction with the State of California and USACE (https://dbw.parks.ca.gov/?page_id=30227 (accessed on 17 March 2016)). This benefits transfer model provides estimates of changes in beach (day-use) value and changes in attendance as beach width changes [17,36]. This model, once state of the practice, has been superseded, but still provides reasonable estimates of how beach recreational (non-market) value changes with beach width. The model was calibrated using data from two beach intercept surveys conducted for the San Diego Area Governments (SANDAG) in 2001 and 2011 [17,36]. Details on the CSBAT model are provided in the Section 2.3.2 [28], and discussion about improvements to the approach are found in the discussion.

In addition to non-market value, Imperial Beach's beach tourism provides substantial tax revenues to the City. The analysis conducted for this study was limited to sales taxes and transient occupancy taxes (TOTs). In addition to measuring economic benefits, this study also measured some tax revenue impacts for the City—notably TOTs and sales taxes. The survey conducted for SANDAG [36] indicates that approximately 25% of Imperial Beach's beach visitors stay overnight and the remaining 75% are day-trippers. Overnight visitors generate more tax revenues and are only included in our estimates of tax revenues. As Imperial Beach builds out its tourist infrastructure, it is expected that the percentage of out-of-town visitors will increase generating more tax revenues to the City. It is also worth noting that Imperial Beach's transient occupancy tax rate, 10%, is low compared to many other coastal cities in California, which charge up to 15%.

The SANDAG survey asked respondents how much they spent on gas, food, lodging, etc., the answers to which were then used to estimate total spending per visitor [29]. For TOTs, total spending on lodging was acquired from the City and applied the 10% TOT rate. For sales taxes, we estimated the spending on items subject to sales tax (e.g., only 30% of grocery sales are typically subject to sales tax).

*2.3. Phase Two: Analysis of Adaptation Alternatives*

The steering committee and stakeholders narrowed a wide range of adaptation alternatives down to five specific strategies assumed to be applied to the entire developed portion of the Imperial Beach shoreline. For each of the five prioritized adaptation strategies, physical changes to both the beach and upland widths were modeled and linked to the economic tradeoffs. As the various widths change over time, upland damages and beach recreation and ecosystem service values change. These adaptation strategies were:

- Coastal armoring
- Beach nourishment
- Living shoreline dune and cobble approach
- Five groins with sand nourishment
- Managed retreat.

2.3.1. Physical Methods

To aid Imperial Beach in identifying the optimal strategy or combination of strategies, the second phase of the case study ran each alternative through a physical response model that projected changes to each upland and beach width over time, driven by accelerated historic erosion rates, based on historical performance and engineering assumptions of performance of each adaptation strategy. For each upland and beach width, the model assigns economic values of development and infrastructure, recreation, and ecosystem services to each width.

The framework compartmentalizes the coastal zone cross-shore into a series of upland and beach widths extending from the surf zone across the beach, and into the upland. A one-line quantified conceptual model tracked physical beach and upland property width changes at 5-year increments for each adaptation strategy (Figure 5). The modeled changes in each width over time were informed from historical observations of nourishment and groin performance [37–39] and clearly stated professional engineering and geomorphic assumptions on the effectiveness of the strategy informed from nearby locations.

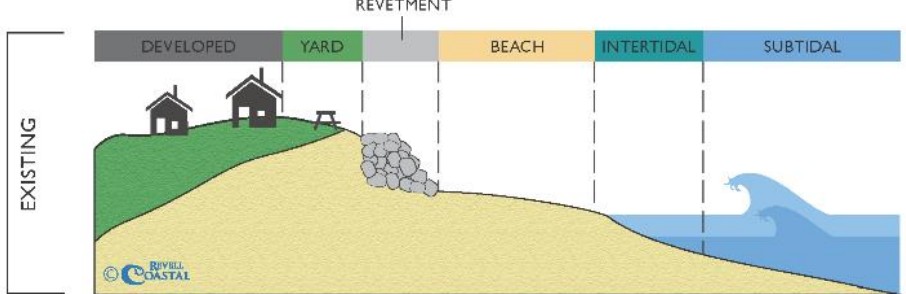

**Figure 5.** Conceptual model of physical upland and beach widths tracked as a result of adaptation strategy implementation over time. For this case study, developed and yard were considered as upland. The revetment in this figure was the adaptation project, while the beach, and intertidal were considered as the beach width.

The numerical and visual representation from the model allowed detailed risk communication to occur with the steering committee that could better evaluate the tradeoffs between the different adaptation alternatives based on their performance, number of treatments, costs, etc. For each width that changes over time, values for fiscal impacts, economic damages, recreational use, and ecosystem function were associated.

To drive the physical response model, historical erosion rates were accelerated based on the assumption of a 2 m sea level rise by 2100. To identify these accelerated erosion rates, historical shoreline change rates were calculated using the Digital Shoreline Analysis System (DSAS), a computer software that calculates shoreline changes by computing rate-of-change statistics developed by USGS [40].

Shoreline reference features were based on a mean high-water elevation of 4.41 feet NAVD88 (La Jolla, California, Station ID: 9410230) and calculated at 33 m spacing along the City (Figure 6). Eight historical shorelines (from 1852, 1887, 1933, 1972, 1998, 2005, 2008, and 2010) were used to calculate a long-term linear regression shoreline change rate.

The long-term shoreline erosion rates were calculated as an average erosion rate of 0.18 m per year, which was used as the historical baseline erosion rate for the study. Future erosion rates were accelerated assuming a linear acceleration from present day to 2100 calculated at 5 year increments to 2 m sea level rise curve to yield projections of future erosion rates (Table 1, blue column) and compared with other published studies by USACE ([26]; Table 1). These USACE rates were a result of analyses completed following a shorter more energetic wave time period between the 1940s and 1980s, which included major storm erosion impacts from the 1982/1983 El Niño.

Based on the historical shoreline change analysis and aerial photography interpretation, two representative beach transects were selected representing typical historical conditions for wide and narrow conditions. One transect was just north of the pier (wide: ~55 m wide), and one transect was near the erosion hotspot near the south end of Seacoast Drive (narrow: ~25 m wide). These transects represent typical minimum and maximum beach widths observed in the historical record and were both used in the model to test the dependency of the framework to physical beach conditions and address one of the uncertainties associated with future physical conditions. Long-term coastal erosion was included in the vulnerability assessment [35]; however, in this conceptual modeling for the adaptation strategy analysis, episodic storm damages were not included given the uncertainty in timing. Major storm wave erosion from historical observations can erode the beach 33 m in any given major storm event [24,26,27]. This limitation is discussed further in the discussion section.

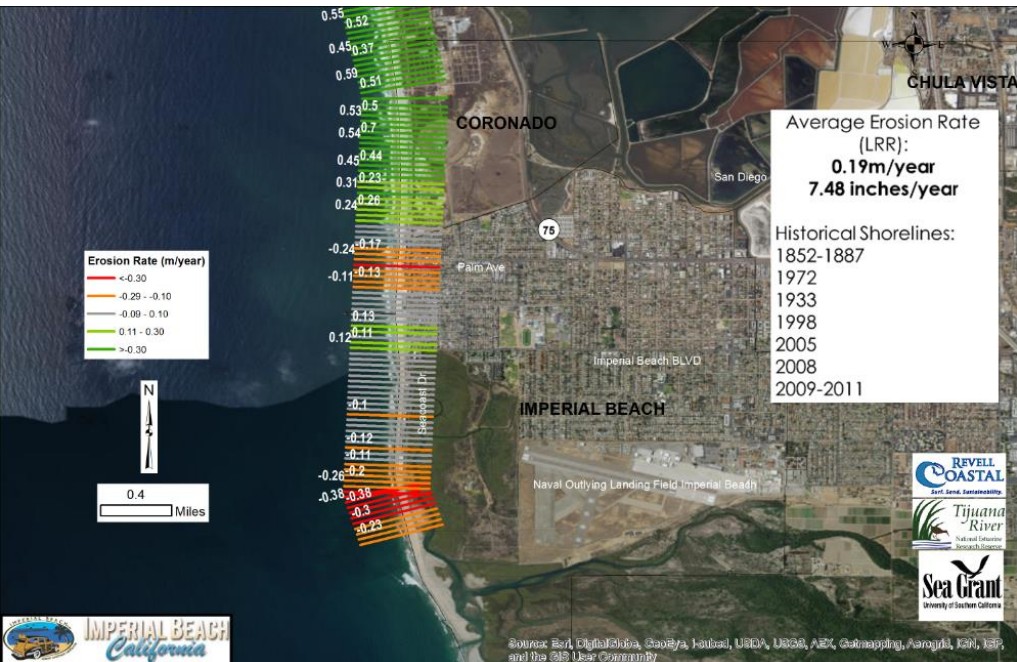

**Figure 6.** Shoreline change rates for 159-year historical time period. Erosion hot spots shown in red. Green is accretion. Grey is within the range of uncertainty.

**Table 1.** Historical erosion rates accelerated over time using a 2.0 m sea level rise curve (m/year).

| Year | This Study | Low USACE | High USACE |
|------|-----------|-----------|------------|
| 2000 | 0.18 | 1.43 | 1.98 |
| 2005 | 0.18 | 1.43 | 1.98 |
| 2010 | 0.27 | 2.10 | 2.93 |
| 2015 | 0.37 | 2.77 | 3.87 |
| 2020 | 0.46 | 3.45 | 4.82 |
| 2025 | 0.48 | 4.12 | 5.76 |
| 2030 | 0.64 | 4.79 | 6.71 |
| 2035 | 0.73 | 5.46 | 7.65 |
| 2040 | 0.82 | 6.13 | 8.57 |
| 2045 | 0.91 | 6.80 | 9.51 |
| 2050 | 1.01 | 7.47 | 10.46 |
| 2048 | 1.10 | 8.14 | 11.40 |
| 2060 | 1.16 | 8.81 | 12.35 |
| 2065 | 1.25 | 9.48 | 13.29 |
| 2070 | 1.34 | 10.15 | 14.18 |
| 2075 | 1.43 | 10.82 | 15.15 |
| 2080 | 1.52 | 11.52 | 16.10 |
| 2085 | 1.62 | 12.20 | 17.04 |
| 2090 | 1.71 | 12.87 | 17.99 |
| 2095 | 1.80 | 13.54 | 18.93 |
| 2100 | 1.89 | 14.18 | 19.85 |

2.3.2. Economic Methods: Cost–Benefit Analysis

To complement the comparison of different physical responses between adaptation approaches, the framework layered on economic analyses associated with the physical changes to quantify the benefits and costs of each alternative. Estimating the costs (e.g., nourishment, and armoring) and benefits (i.e., losses averted, recreation, and ecosystem services) of adaptation is challenging given the limitations of our scientific knowledge about coastal processes and economic valuation of ecosystem services. The CBA framework employed in this study considers:

1.  Estimating the benefits of each adaptation alternative. The primary benefits measured in this study were:
    a.  Flood damage avoidance to private and public property;
    b.  Recreation (non-market) benefits of beaches;
    c.  Ecological value of beaches (measured at replacement cost).
2.  Estimating the project lifecycle costs of each alternative including, construction, maintenance, and (potentially) removal of structures.
3.  Calculating net present value of costs and benefits: the difference between the present value of cash inflows (benefits from adaptation) and the present value of cash outflows (cost and maintenance throughout the implementation period). Net present value is calculated as follows:

$$Net\ Benefits = \sum_{t=0}^{n} \frac{Bt - C_t}{(1+r)^t}$$

where
$B$ represents benefits of the adaptation strategy;
$C$ represents costs of the adaptation strategy;
$t$ represents time from 2015 to 2100;
$r$ is the discount rate (discussed in more detail below).

The benefit–cost analysis incorporated the following inputs: (1) losses to public property and infrastructure; (2) losses to private property (land and structures); (3) changes in beach recreation; (4) changes in ecosystem services; and (5) project lifecycle cost estimates

of adaptation strategies. In addition, the study estimated the tax revenue impacts of each strategy, but these were not incorporated into the CBA framework.

Private Property

The existing coastal flooding and erosion risk to private property was based on parcel-specific assessor's property tax database provided by the County. These "parcel data" contain detailed geospatial information about the location and size of the parcel, the size of the structure, the type of structure, (e.g., single family dwelling, multiple family dwelling), and the elevation of the parcel. The parcel data also contain information on the assessed value of the property—the value placed on the property for tax purposes. California's Proposition 13 limits increases in assessed value to 2% per year, implying assessed valuations on older properties (when housing inflation was >2%) are significantly lower than market or replacement costs. To correct for this inaccuracy, we developed a housing price index based on averages of local housing price increases from Zillow and the Case-Shiller Real Estate Index for San Diego, which contains housing inflation rates for the San Diego area [41,42]. This method was applied to properly value each parcel from assessed value to current market rates. (The housing price index is similar to the consumer price index, in that it estimates the average inflation rate for housing in a particular area. Because parcel data also contains the date of the last sale for a property, one can compare the current assessed value to an estimated fair market value today.) The structures on the parcels (e.g., houses) were then valued using standard USACE and Federal Emergency Management Agency (FEMA) flood damage estimation techniques [9,30]. These techniques value a structure by size, number of stories, type of dwelling, and type of construction [43] and the damages are estimated depending upon the flood depth/duration.

Public Property

Public property such as schools, libraries, and other buildings owned by various government entities is not subject to property tax. In these cases, parcel data do not contain assessed value and often contain little information other than the size of the parcel; therefore, a standard rate was applied based on parcel size. The City provided detailed information on the replacement cost of these structures, with the exception of properties owned by the U.S. Department of Defense.

Infrastructure

The two most important types of physical infrastructure at risk estimated in this project were roads and water pumps. It was assumed that all roads/infrastructure would need to be replaced when threatened by erosion, and these assets were valued at the estimated replacement cost at the time of failure. The team determined the timeline where replacement should occur and identified "trigger points" to begin adaptation planning so that planning, permitting, financing and implementation could occur before potential vulnerabilities were realized. However, the analysis did not include the additional costs of acquiring new sites for relocating.

Recreation

As mentioned in Section 2.2, the study applied an economic valuation model developed for the State of California and USACE, the CSBAT, which allows one to estimate the gain or loss in recreational value as beach width decreases (e.g., due to erosion) or increases (e.g., due to nourishment) [17,36]. This model was calibrated using survey data collected at San Diego area beaches [29]. The maximum non-market value in CSBAT was determined by results from the Southern California Beach Project, which was based on a comprehensive Random Utility Model of beaches in southern California [44]. The "day-use" values for a day at the beach in this study were relatively low, in the range of USD 10 a day, due to the ease of substitution. In the CSBAT model, the maximum day-use value is USD 16, in 2015 dollars, but narrower beaches provided lower day-use value.

Our analysis also included estimates of increased TOTs and sales tax revenues generated locally (all TOT revenues go to the City, but only a small portion of sales taxes do). As with similar studies, taxes were not incorporated into the formal benefit–cost framework but were reported separately [45,46].

Ecological Benefits of Coastal Habitat

The State of California has committed itself to more nature-based solutions when adapting to sea level rise [2]. Thus, the need to value ecosystem services is important to integrate into CBAs related to adaptation planning. Unlike previous studies discussed above, this study aims to comprehensively value beaches, including *all* of their ecosystem services. This framework approach considers typically valued ecosystem services of storm buffering and recreation as well as specifically attempting to incorporate the additional direct and indirect use values associated with ecosystem services [3].

Our analysis explicitly incorporates the recreational value of beaches through a benefit transfer model and dissipation benefits at least partially by estimating erosion and flood losses/damages [47]. However, these benefits constitute only two of the 14 benefits listed by Defeo et al. [3] in Table 2.

**Table 2.** Sandy Beach Ecosystems Services by Use Value Type [3].

| Sandy Beach Ecosystem Services | Direct Use Value | Indirect Use Value |
|---|:---:|:---:|
| Sediment storage and transport | | X |
| Wave dissipation and associated buffering against extreme events | | X |
| Dynamic response to sea-level rise (within limits) | | X |
| Breakdown of organic materials and pollutants | | X |
| Water filtration and purification | | X |
| Nutrient mineralization and recycling | | X |
| Water storage in dune aquifers and seawater discharge through beaches | | X |
| Maintenance of biodiversity and genetic resources | X | |
| Nursery areas for juvenile fishes | X | |
| Nesting sites for turtles and shorebirds, and rookeries for pinnipeds | X | |
| Prey resources for birds, fishes, and terrestrial wildlife | X | |
| Scenic vistas and recreational opportunities | X | |
| Bait and food organisms | X | |
| Functional links between terrestrial and marine environments on the coast | X | |

Quantifying these benefits is challenging [1,4]. Quantifying each ecological service with a definitive value is beyond the scope of this study and will require substantial future research because of the interconnected reality of an ecological system. However, it is clear that the ecological value of these beaches beyond recreation and storm buffering is significant. This study proposes a framework for including recreation and ecosystem services as different parts of a whole, integrated system with associated values to inform adaptation.

Because ecosystem services represent a flow per year, these estimates of a total value per acre were converted to an annual flow of services. After discussions with the steering committee, the team used a very conservative value for the flow of these additional ecological services of USD 30,000 per acre per year. This estimate is consistent with Raheem et al. [48], although their estimates, like most others, leave out most ecological services other than recreation and storm buffering.

In other areas of ecological mitigation/restoration, scientists and policy makers have recognized our lack of understanding of these ecosystem services and have applied a precautionary principle [43] in order to ensure the preservation of natural habitat. This precautionary principle recognizes that we cannot value some ecosystem services precisely. In the case of wetlands, an offset approach is used where wetland (acres) lost to development are replaced, at a cost. Typically, an offset ratio is larger than one because newly restored wetlands may have lower ecological services than an established wetland [1]. If one applies a similar approach to beaches in California [47,49], one finds that the cost of replacing an entire beach ecosystem is approximately USD 5.9 million per acre [47]. The monetary value of USD 30,000 represents only 0.5% of USD 5.9 million, so our assumption assumes a "return" of half of one percent per year on all ecosystem services other than recreation or storm buffering. Similarly, if one examines local real estate values [50] the average value of coastal real estate (land not structures) in 2014 was USD 1.25 million; USD 30,000 represents only 2.4% of this value.

While not a perfect comparison, the California Department of Fish and Wildlife conducted a natural resource damage assessment for ecological losses associated with the cleanup of a small oil spill at Refugio Beach in Santa Barbara County. This assessment estimated the value of ecological losses to habitats and species [51]. Excluding human recreational uses or storm prevention, the cleanup costs of these damages to southern California beach habitats totaled USD 18.1 million over a slightly larger stretch of coastline. This was not a restoration cost, but only a cleanup cost, and although these beaches and coastal habitat are distinct, they indicate the value of California's coast and some of the ecosystem service values [46].

Ecological Impact of Nourishment

Although nourishment projects enhance beach width and hence recreation, the impacts of nourishment on beach ecology are mixed. Typically, nourishment projects involve pumping sand onto a beach and (typically) bulldozing the sand in place that disrupts the foraging, nesting functions of beaches and buries many creatures who live in the sand and form an important part of the sandy beach ecosystem (e.g., sand crabs). The result is generally a significant loss of natural beach ecosystem function for a time. Numerous studies [3,4,18,52,53] have found detrimental environmental impacts from nourishment. To account for nourishment impacts, this study assumed that the value of ecosystem services would be diminished by 50% in the first year after nourishment and gradually recover at a rate of 15% a year until full ecological capacity is reached (typically in 5–7 years) [38,53]. While this assumption is clearly an oversimplification, it captures the loss in ecosystem functioning far better than assuming (as is typical) that beach ecosystem services, beyond recreation and protection of upland property, have no value.

Engineering Cost Estimates

Each of the adaptation strategies required engineering costs for construction, replacement, removal, and maintenance. All lifecycle cost estimates were provided by an experienced licensed coastal engineer with more than 30 years of experience estimating construction and restoration costs. These lifecycle estimates were based on professional judgment after reviewing similar regional projects [29], the City's Capital Improvement Plan, and other project cost estimates provided by steering committee members.

During the steering committee engagement process, decisions about the five prioritized approaches for detailed analysis were extensive. In response to this engagement, the study team did an initial evaluation of elevating the structures at risk. This accommodation approach, ultimately was not one of the top five selected preferences for detailed analyses but the cost differences proved to be the deciding factor and deserve mention. The elevating and flood-proofing costs for existing structures was calculated based on available structural footprint data from the City using GIS and estimated retrofit costs to elevate existing structures at USD 250 per square foot for structures (Table 3).

The total retrofit cost of elevating existing structures was quite high, ranging from USD 198 million in 2047 to USD 385 million in 2100. However, our analysis of the total potential damages from tidal and event flooding was significantly lower (less than USD 100 million) in all time horizons, indicating that retrofitting/elevating existing structures was not cost-effective in this setting. However, there are a variety of policy approaches to improve new construction or redevelopment standards through building codes to encourage increased elevation of structures in hazardous areas over time that were recommended for inclusion in future updates to City planning documents [54].

**Table 3.** Estimated retrofit costs of elevating structures in flood zones.

| Sea Level Rise | Elevated Sq Ft | Total Cost |
| --- | --- | --- |
| 2047 (0.5 m) | 791,630 | USD 197,907,500 |
| 2069 (1.0 m) | 1,039,031 | USD 259,757,750 |
| 2100 (2.0 m) | 1,539,025 | USD 384,756,250 |

The steering committee and stakeholders identified five specific strategies applied to the entire developed portion of the Imperial Beach shoreline. The construction costs and maintenance estimates used for the detailed adaptation strategy analysis included seawall removal and construction, groin construction, cobble, and dune construction, as well as the costs of removal for managed retreat scenarios (Table 4). The framework analysis also accounts for the fact that maintenance costs for armoring solutions will increase as beach widths narrow, creating more frequent wave energy impacting the armoring structures.

**Table 4.** Engineering construction and maintenance cost estimate break down by unit costs. Source Joe Ellis, P.E., Marathon Construction.

| Structure | Initial Cost (USD)/Unit | Units | Total Cost (USD) | Maintenance |
| --- | --- | --- | --- | --- |
| Seawall Removal | $1000 per ft | 7920 ft. | $7,920,000 | |
| Nourishment (30 m × 2.6 km) | $20 per cy ($1.1 M/acre) | 1 million cy 2.6 (acres) | $20,000,000 | (add USD 1/year) per cy |
| New Seawall | $4500 per ft | 7920 ft | $35,640,000 | 2% every 10 years |
| New Groin | $4000 per ft. (930ft each × 4 additional) | 3720 ft | $14,880,000 | 5% every 10 years |
| Cobble | $3000 per ft | 7920 ft | $23,760,000 | |
| Dune Sand | $1000 per ft | 7920 ft | $7,920,000 | |
| House Removal | $10 per sq ft | | | |
| Condo Removal | $20 per sq ft | | | |
| Road Removal | $4 per sq ft | | | |
| Pipe Removal | $20 per ft | | | |
| Pump Station Removal | $200 per sq ft | | | |
| Dune Restoration | $77,000 per acre | 11.8 acres | $910,000 | |

Tax Revenue Impacts

The economic impacts/spending discussed in the previous section also implies increased tax revenues. Our analysis focused on two primary sources of local tax revenues that would be influenced by the adaptation strategies discussed in this study: sales taxes

and TOTs. Our analysis relied on recent survey data [9,37] and local estimates of beach attendance to estimate spending and taxes. To simplify, we assumed that attendance increased with the population growth and that the demand for beach recreation in Imperial Beach has an income elasticity of one—that is, SANDAG's projection for annual increase in household incomes within Imperial Beach of 0.7% will increase beach visits by 0.7%, annually. Similarly, the analysis assumes that demand for beach attendance will grow with the population of San Diego County which according to the California Department of Finance demographic unit, will average 0.3% per year.

*2.4. Comparison of Adaptation Strategies Using Net Benefits*

Results of the economic analysis are reported as net benefits, which are the total benefits provided by a particular adaptation option, projected over time, minus the lifecycle costs of implementing and maintaining the project (largely engineering costs). Although not part of the net benefit estimates, policy makers are often interested in the economic impacts of these measures. Economists measure economic impacts by measuring changes in spending for various strategies.

Our analysis incorporates estimated changes in direct beach recreational spending with various adaptation strategies. As discussed above, beach visitation varies with beach width in our modeling. Consequently, strategies that maintain or enhance beach width, such as nourishment, create positive economic benefits and generate more local tax revenues. On the other hand, armoring strategies often lead to diminished beach width, which not only lowers the non-market recreational benefits that visitors get from a day at the beach, but also lowers total spending in Imperial Beach, because fewer visitors go to narrower beaches.

When considering benefits and costs incurred over a number of years, dollar values must be adjusted to reflect the fact that a dollar received today is considered more valuable than a dollar received in the future, because a dollar received today could be invested elsewhere to produce additional wealth. This adjustment is known as the discount rate. To make this adjustment, it is important to select a discount rate that will account for the diminishing value of benefits received in the future. On the other hand, the use of a market rate, even a low one like 3%, implies that future generations are less valuable.

Given the potentially enormous costs of climate change to future generations and the longer time scale, many environmental economists have proposed applying lower discount rates when analyzing the economic impacts of climate change [7,42,55]. One of the most widely cited reports, the Stern report [39], applied a 1.4% discount rate. Arrow et al. [7] indicate that climate change modeling presents a unique set of issues given the uncertainty involved and the potential for catastrophic outcomes, even if the probability of such outcomes is low. Weitzman [30] recommends a 1% discount rate for periods exceeding 75 years and 0% for periods exceeding 300 years.

This study followed the approach of Weitzman [30] and applied a 1% discount rate, which implies that consumption and expenditures by future generations are relatively more important than implied by a higher discount rate. However, our sensitivity analysis, discussed later, examined benefits and costs using differing discount rates and demonstrated the robustness of these findings.

## 3. Results

The results of this study include both the baseline initial vulnerability assessment and the evaluation of the five adaptation alternatives under the conceptual framework. First, the physical vulnerabilities identified potential impacts to various sectors in the vulnerability assessment. Second, for each adaptation strategy, the physical results of the upland versus beach width change model and the associated economic impacts for the five adaptation strategies are presented. Third, a comparison is presented of the net benefits of these alternatives. Crucially, this comparison also involved consideration of the ecological cost of various adaptation strategies.

### 3.1. Vulnerability Assessment

The vulnerability assessment results from the geospatial analyses showed potential impacts to all examined coastal hazards, including coastal erosion, coastal wave flooding, and tidal inundation for the variety of sea level rise scenarios (0.5, 1.0, 1.5, and 2.0 m). All beach accesses and oceanfront properties are in existing coastal erosion and coastal flood hazard zones associated with a 100-year wave event. These results assume a do-nothing approach with no adaptation strategies implemented representing a worst-case scenario.

Coastal erosion will likely accelerate historical erosion rates as sea level rises. Accelerating historical erosion rates based on 2.0 m sea level rise projects an escalation of erosion from 0.18 m per year currently to 1.9 m per year with 2.0 m of sea level rise.

Stormwater and nuisance flooding associated with high tides will increase in frequency and duration as tidal elevations decrease the stormwater conveyance capacity. Tidal inundation poses little risk under existing conditions, but impacts escalate dramatically between 1.0 and 2.0 m of sea level rise.

The key vulnerabilities across the City fell into three sectors in four neighborhoods—south Sea Coast Drive, north of Palm Avenue/Carnation Avenue, neighborhood around Bayside Elementary, and Seaside Point (Figure 2). The three sectors most at risk at 2 m of sea level rise are:

- Stormwater—substantial decrease in stormwater capacity;
- Land Use—30% of all parcels and buildings;
- Roads—40% of all roads impacted.

Vulnerabilities due to exposure to each coastal hazard were examined to show potential economic impacts to the residential land uses and private properties (Figure 7). Results show that coastal erosion causes the most economic damages due to loss of land and structure damages, while storm related wave flooding is second. In addition, another form of flooding, tidal inundation, becomes a significant economic issue between 1 and 2 m. Currently, this flooding is often considered a nuisance, and these damages are often overlooked in sea level rise studies. However, tidal flooding will create significant economic damages to property in Imperial Beach, without an adequate adaptation strategy. Many properties damaged will be inland, not directly on the coast, because Imperial Beach's topography includes some low-lying inland areas—and our study area included these low-lying areas away from the coast. According to the U.S. Census [56], Imperial Beach has nearly twice the number of households living in poverty (18.9%) as California overall (11.8%). Many of the homes experiencing tidal flooding in Imperial Beach will not be expensive homes by the beach, but homes with working families living from paycheck to paycheck. The economic damages of tidal flooding jump substantially between 1 and 2 m of sea level rise.

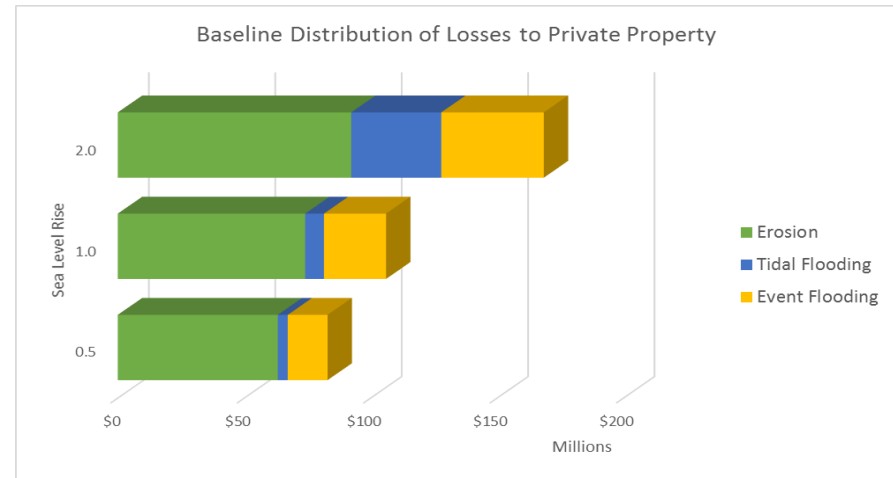

**Figure 7.** Losses and damages for private development from each of the coastal hazards through time.

*3.2. Evaluation of Framework to Adaptation Alternatives*

This section provides a description of results of the framework application to each of the five selected adaptation strategies considered in the study: coastal armoring, beach nourishment, living shoreline (dune and cobble approach), five groins with sand nourishment, and managed retreat. For each strategy we provide results from the physical analysis of modeled beach width and impacts to upland property, economic considerations, and ecosystem services.

### 3.2.1. Coastal Armoring

The coastal armoring strategy considered the protection of upland property by relying on the existing mix of coastal armoring structures through 2030. In 2030, the coastal armoring structures were assumed to be upgraded to a uniform vertical recurved seawall with a narrower footprint on the public beach. This assumption of the footprint allowed an increase of 20 feet of beach as the revetment was replaced by a recurved seawall. As sea level rise and continuing erosion occur, the beach has no room to migrate inland and eventually disappears; thus, while the upland remains protected, the beach is lost. The implementation and evolution of this strategy can be seen in Figure 8. In Imperial Beach, the current coastal armoring policy requires that any new or substantive repairs to coastal armoring be in the form of a vertical seawall located on the private property.

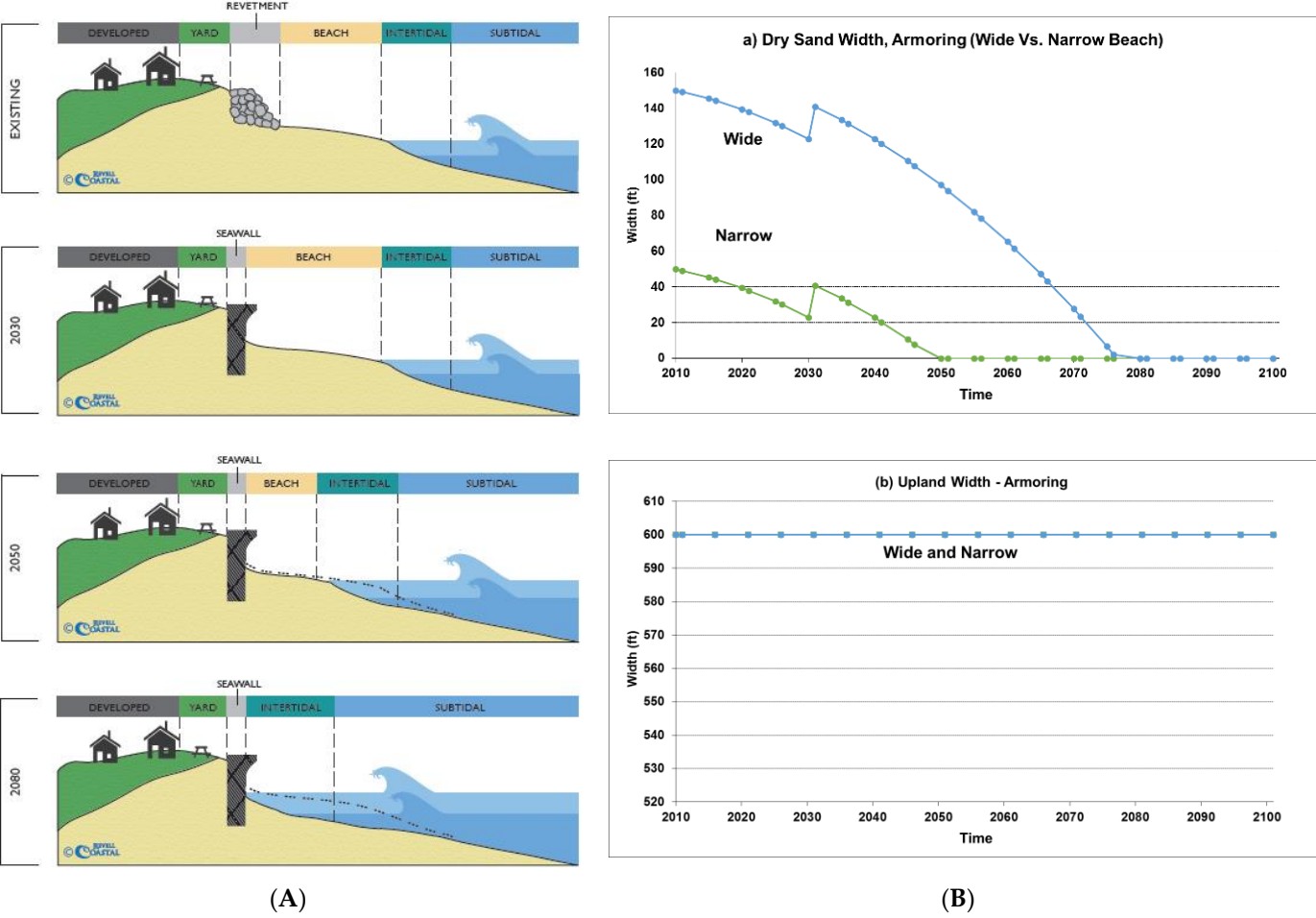

**Figure 8. (A)** Evolution of the armoring adaptation strategy, showing a small increase in usable beach as the existing revetments are replaced with vertical seawalls, then the loss of dry and eventually intertidal sand beaches over time. **(B)** Dry sand beach width over time with armoring (wide vs. narrow beach) (Ba top plot); upland width over time with armoring (wide vs. narrow beach) (Bb bottom plot).

Conceptual Upland and Beach Response Model

Results from the physical analysis of beach width versus upland property show that under both beach width conditions (wide and narrow), upland property would be maintained into the future, while the beach is eventually lost (Figure 8b). In 2030, when the 8 m revetment is replaced with a 1.5 m seawall, 6.5 m of beach width is recovered.

For the narrow beach condition, dry sand beaches would likely completely disappear by 2050, and damp sand beaches (those that are only accessible at low tide) would disappear by 2060. For the wide condition, the dry sand beach disappears by 2065, and the damp sand beach disappears by 2075.

Economic and Ecological Considerations

The armoring adaptation scenario involves the replacement of the existing revetment with a recurved seawall, projected to cost approximately USD 44 million initially. In the armoring scenario, private properly and public property on the coast are protected, resulting in a minimal projected loss to property. However, as the beach is squeezed between the ocean and the seawall, armoring imposed high ecological costs by increasing beach erosion, which limits beach width and habitat area and interfering with many of the natural processes, such as allowing the beach to migrate inland, and some fauna to retreat during coastal storms [57,58]. Using replacement costs, these losses are conservatively estimated to result in only USD 6 million (narrow) in preserved ecological value by 2100. Finally, although replacing the revetment leads to an initial increase in beach width, the beach will eventually erode in front of the seawall, resulting in comparatively minimal recreational value of USD 100 million (narrow) to USD 400 million (wide) by 2100.

3.2.2. Beach Nourishment

The beach nourishment alternative was selected to emulate what has been the most common historical practice in Imperial Beach, namely, to periodically nourish the beaches while maintaining the existing structures. The intent of this alternative is to protect the existing upland and maintain a beach. The implementation and evolution of this strategy can be seen in Figure 9.

Conceptual Upland and Beach Response Model

Results from the physical analysis of nourishment, the beach width versus upland property, show that with enough nourishment placements, the upland area can be protected while a sandy beach is maintained. Assumptions included that nourishment size was similar to historical nourishment volumes of about a million cubic yards (~33 m by 2.6 km) [26,29]. Once nourished, the sand volume and related beach widths decreased by 50% every 5 years following placement as it equilibrated to the natural coastal profile. Each renourishment was triggered before upland property damages would occur.

To maintain a recreational beach to accommodate 2 m of sea level rise, model results project that between 9 (wide) and 11 (narrow) nourishments would be required by 2100 to maintain beach width and protect upland property (Figure 9b). As sea level rises and erosion rates increase, the frequency of nourishment increases. In the near future, nourishments tend to occur every 15 years or so, but by 2100, it was projected that the nourishment cycle would have to occur about every 5 years.

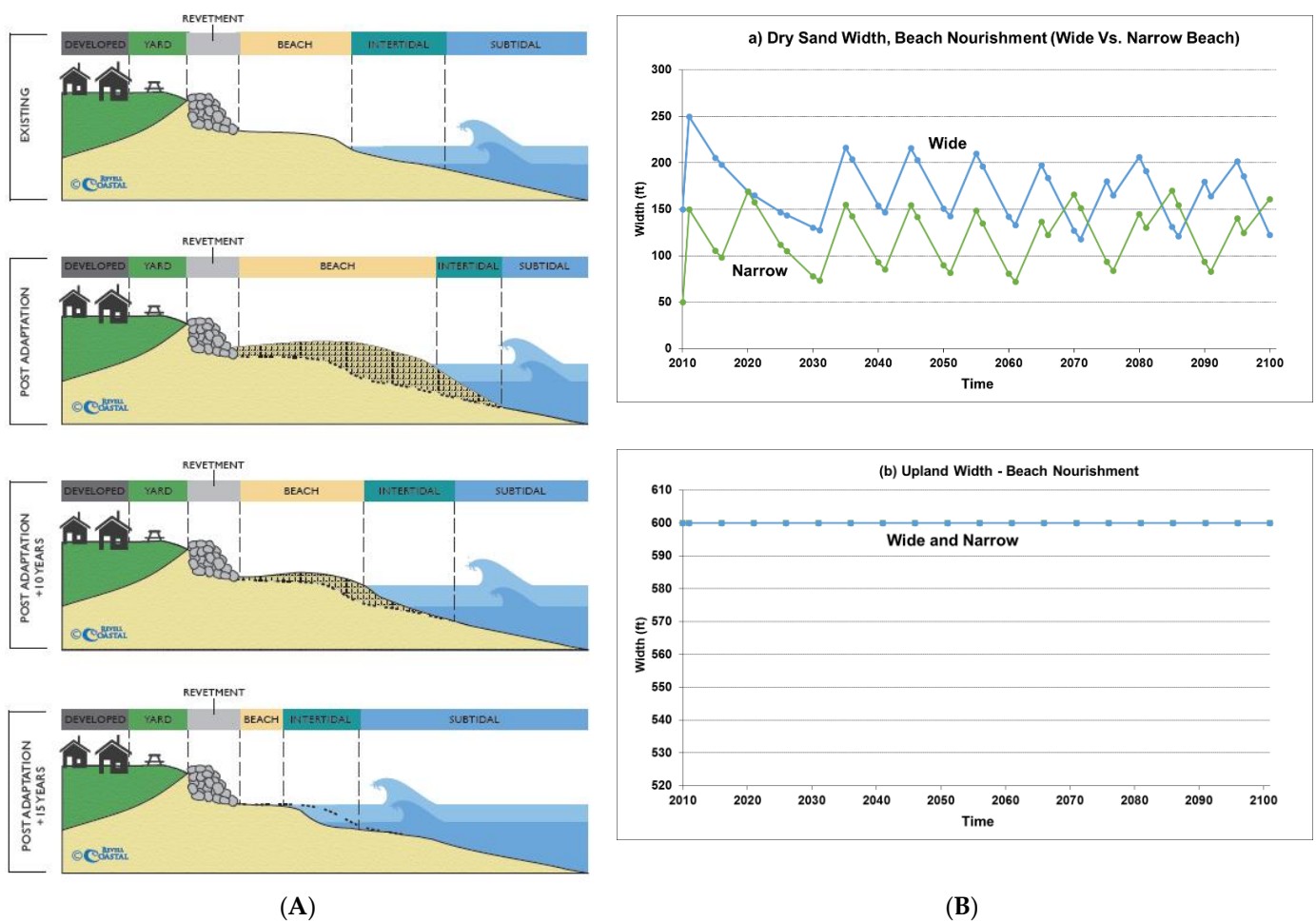

**Figure 9.** (**A**) Evolution of the beach nourishment strategy over time. Example does not illustrate the shortening time between nourishment cycles as sea level rises. (**B**) Dry sand beach width over time with armoring (wide vs. narrow beach) (Ba top plot); upland width over time with armoring (wide vs. narrow beach) (Bb bottom plot).

Economic and Ecological Considerations

Our results indicate that beach nourishment could provide an effective adaptation strategy over time because much of the beach value is preserved while upland damages are avoided. Based on the literature [38,53] and consultation with beach ecologists, it was assumed that beach ecosystem values diminish by 50% after nourishment but recover linearly over 7 years. However, given the increasing construction cost and frequency over time, the ecological values start to diminish and disappear once the nourishment cycles drop below 7 years.

Overall, nourishments were effective in protecting property and preserving recreational value, with fewer negative impacts to ecology. The nourishment cycles as the sand erodes create a pattern of overall increasing recreational value over time, however decreasing value between nourishment cycles as the beach shrinks. Overall, the beach was projected to provide recreational value of USD 593 million (narrow) to USD 722 million (wide). Given the initial cost of nourishment is projected at USD 20 million, this strategy as modeled is highly cost-effective. Our results indicate that beach nourishment could provide an effective adaptation strategy over time because much of the beach value is preserved while upland damages are avoided.

### 3.2.3. Living Shoreline Dune and Cobble Approach

The living shoreline dune and cobble approach alternative represents a living shoreline approach and attempts to emulate what was likely the natural form and function

of the coastal landscape prior to substantial human influence and development. The historical ecology of the study area contained natural dunes and wide sandy beaches underlain by cobble, thus, this alternative is a nature based green protection or "living shoreline" approach.

The intent of this alternative is to protect the existing upland with a combination of beach sand nourishment, cobble placement, and dune creation. The resulting strategy allows erosion of the nourished sandy beach based on historical nourishment volumes [26]. Eventually the sandy beach erosion exposes the cobbles triggering a reduced rate of erosion. As the cobbles narrow, dune erosion starts to increase. Once the dune erodes by one-third, with the crest elevation still intact, another living shoreline/hybrid dune is assumed to be implemented to avoid upland damages. The implementation and conceptual evolution of this strategy can be seen in Figure 10.

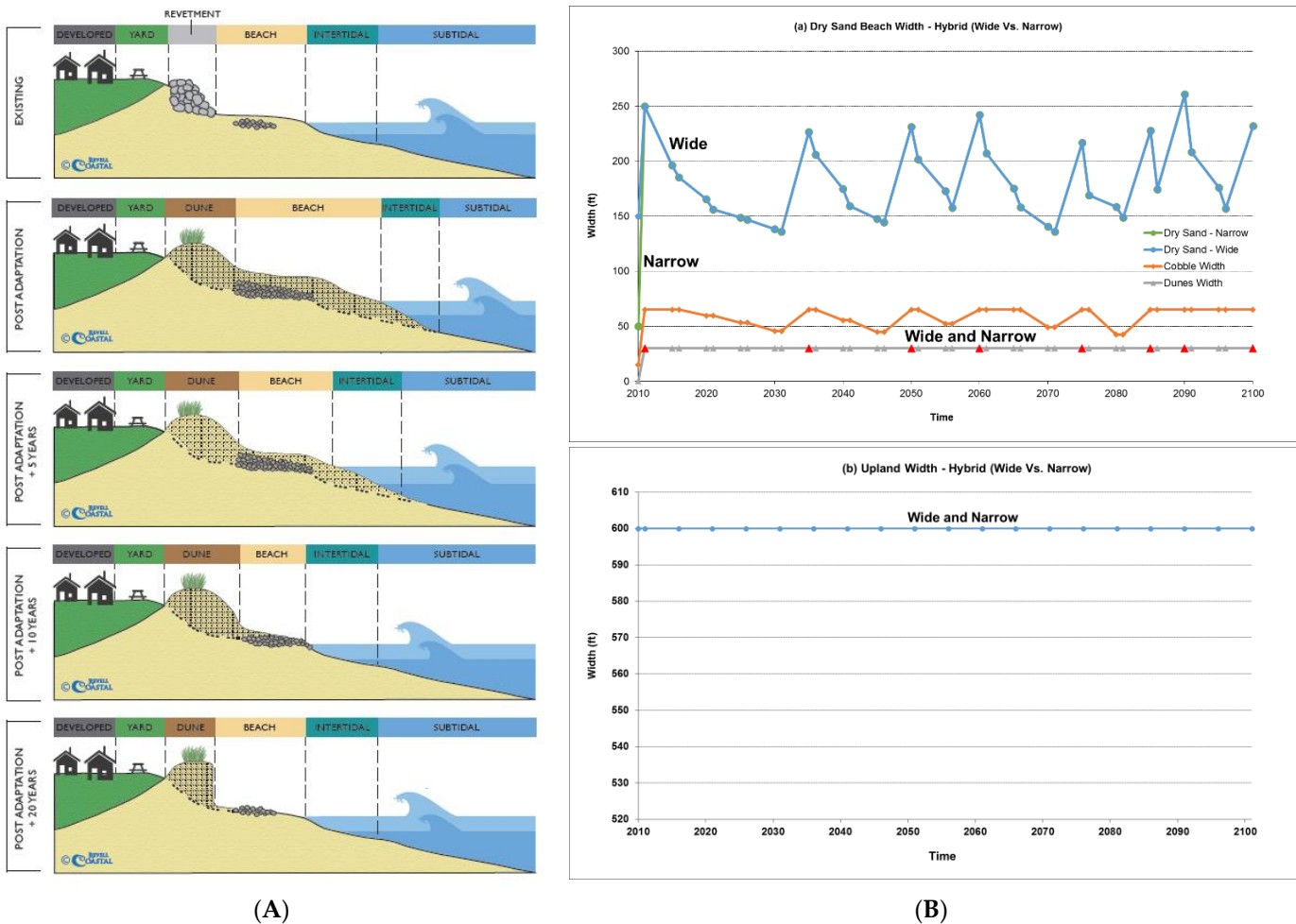

**Figure 10.** (**A**) Evolution of the living cobble and dune shoreline strategy over time. Example does not illustrate the shortening time between nourishment cycles as sea level rises. (**B**) Dry sand beach width over time with armoring (wide vs. narrow beach) (Ba top plot); upland width over time with armoring (wide vs. narrow beach) (Bb bottom plot).

Conceptual Upland and Beach Response Model

Results from the physical analysis of beach width versus upland property show that the upland can be protected while maintaining a sandy beach with enough hybrid dune nourishment placements. To maintain a recreational beach to accommodate 6.5 feet of sea level rise, model results project eight (wide beach) and nine (narrow beach) reconstruction treatments by 2100 to maintain beach width and protect upland property.

As sea level rises and erosion rates increase, the frequency of the hybrid nourishment placements increases. In the near future, nourishments tend to occur every 15 years or so, but by the end of century, it is projected that the nourishment cycle would have to occur about every 5 to 10 years (Figure 10b).

Economic and Ecological Considerations

The economic considerations here are similar to beach nourishment, above, as both endeavor to preserve the beach width for habitat and recreation. However, the cobble inclusion in the living shoreline approach provides additional resilience reducing the reconstruction cycles and maintaining ecosystem services farther into the future. In the future, a more detailed analysis may want to consider whether a living shoreline dune and cobble restoration has more ecological value. The living shoreline approach has higher initial costs of USD 52 million. More in-depth analysis of the specific ecological benefits of this strategy may offset that cost with benefits to the ecosystem, particularly given the location of Imperial Beach to the nearby Tijuana River and Slough. Similar to nourishment, the nature of the dune replenishment created a cyclical pattern of gradually increasing beach attendance and associated value (USD 893 million in 2100). Our estimates indicate a value of USD 58 million in ecological benefits.

### 3.2.4. Five Groins with Sand Nourishment

The intent of this adaptation alternative is to protect the existing upland using a series of five cross-shore groins to impound sand and widen and maintain the beach. The adaptation alternative was based on the original USACE concept [8], with some modifications—an extension of the existing two groins and construction of three new groins to a length of ~300 m. Revisions to the original USACE concepts were based on more recent work at nearby Oceanside, California, which identified a longer length of the groins and by SANDAG, which evaluated the alongshore spacing for sand retention [17].

The construction of the groins as a sediment retention structure is coupled with a nourishment that fills (or charges) the groin beach compartments and reduces the likelihood of downcoast erosion impacts to the adjacent City of Coronado and Silver Strand State Beaches. This charging of the groin field is akin to filling up a leaky barrel with sand intended to mitigate downcoast erosion commonly associated with groins. This alternative assumes that existing coastal armoring remains, and the widened beaches reduce the armoring maintenance costs.

The resulting strategy constructs the groins and nourishment, with additional nourishment triggered once the beach reaches a certain threshold beach width defined as the preexisting beach width. Additional implementations and the evolution of this strategy can be seen in Figure 11.

Conceptual Upland and Beach Response Model

Results from the physical analysis of beach width versus upland property show that the upland and sandy beach can be protected by implementing a groin adaptation strategy. To maintain a recreational beach to accommodate 2 m of sea level rise, model results project six (wide) or seven (narrow) nourishment placements along with groin maintenance by 2100 to maintain beach width and protect upland property (Figure 11b).

As sea level rises and erosion rates increase, the frequency of the nourishment placements increases. In the near future, nourishments tend to occur every 25 years or so, but by the end of century, it is projected that the nourishment cycle would have to occur about every 10 years along with increased maintenance of the groins.

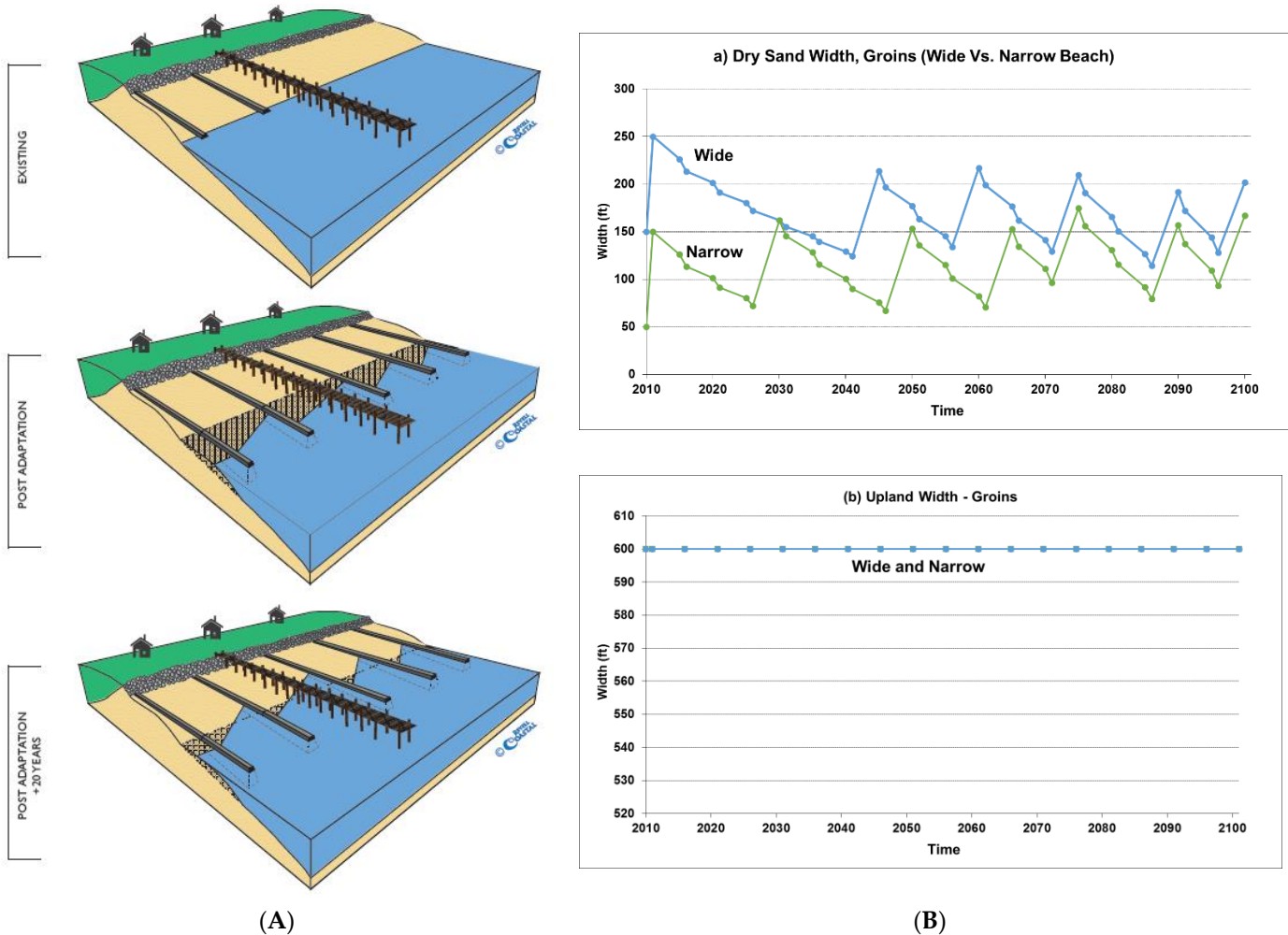

**Figure 11.** (**A**) Sand retention with groins adaptation strategy over time. (**B**) Dry sand beach width over time with armoring (wide vs. narrow beach) (Ba top plot); upland width over time with armoring (wide vs. narrow beach) (Bb bottom plot).

Economic and Ecological Considerations

Overall, the groin approach maintains beach recreation and ecosystem services longer into the future and thus provides higher benefits into the future. In this study, we assumed that the groin structures themselves did not have a positive or negative impact on recreational usage. Our analysis of the recreational and ecological impacts was limited by the scope of the project to where the specific nuances of this adaptation were not fully modeled. For example, groins can increase the recreational experience for fishermen and possibly surfers but can also detract from the aesthetics of a beach. Groins do inhibit the movement of some biota on the beach, but this was also not factored into the ecological value generated by our analysis of between USD 31 million (narrow) and USD 52 million (wide) by 2100. Because of the nourishments included with the groins, overall recreational value was preserved in a similar cyclical manner, for a USD 797 million value in 2100 (wide; narrow scenario USD 618 million) compared to the USD 35 million initial cost of the project.

### 3.2.5. Managed Retreat

Managed retreat can be implemented in a variety of ways, most of them controversial pitting public trust resources versus wealthy private property interests [5,6,16]. This managed retreat adaptation alternative prioritizes preservation of the beach and its associated recreation and ecological benefits above upland property protection. This adaptation approach assumes removal of the shoreline armoring in 2030, allowing the coast to transgress,

eroding inland and vertically in elevation. As buildings and infrastructure are damaged, there are associated removal costs. The implementation and evolution of this strategy can be seen in Figure 12.

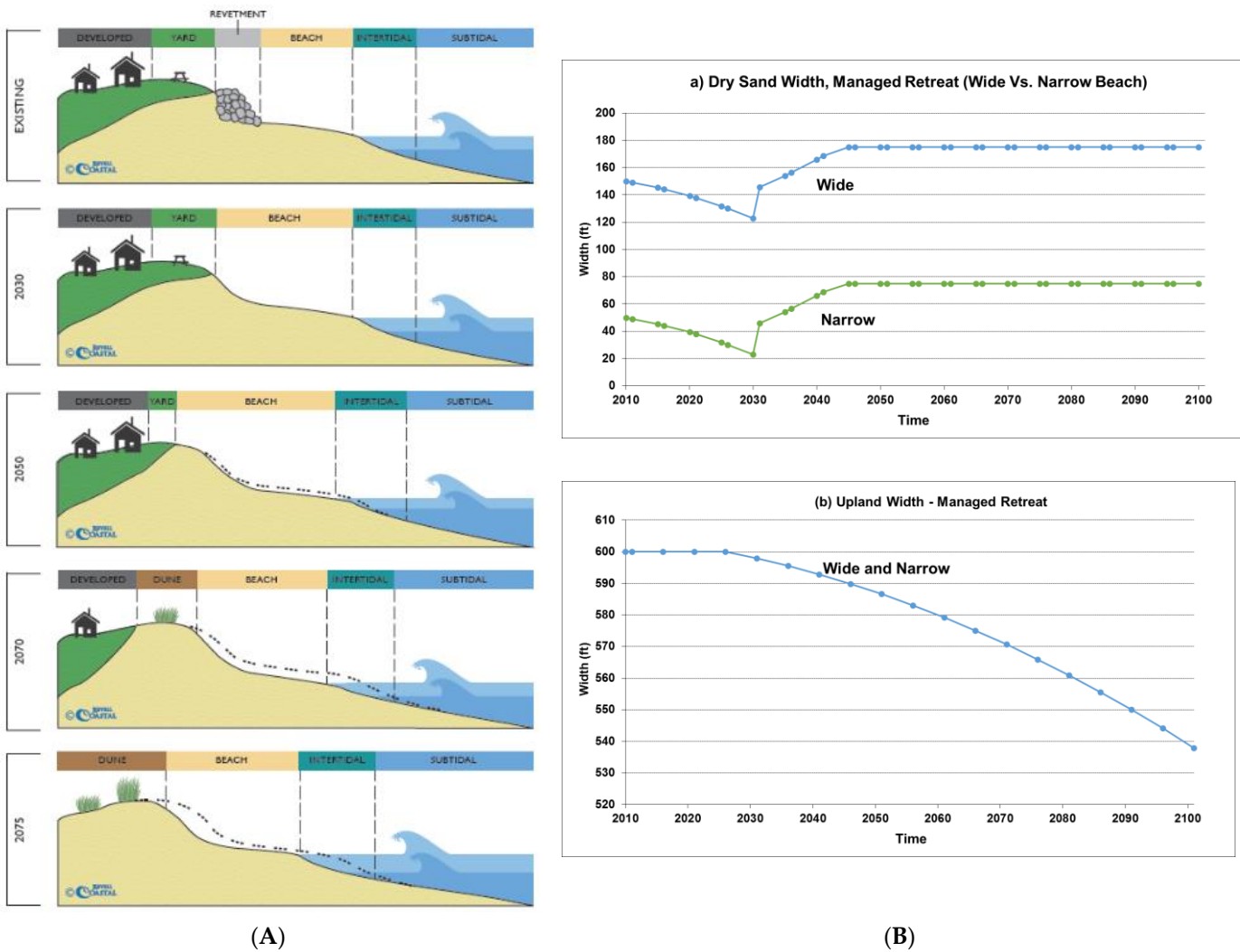

**(A)**                                                                                                          **(B)**

**Figure 12.** (**A**) Managed retreat adaptation follows removal of existing coastal armoring in 2030, with coastal erosion damaging private property while allowing beaches to migrate inland and maintain their public recreation and natural values. (**B**) Dry sand beach width over time with armoring (wide vs. narrow beach) (Ba top plot); upland width over time with armoring (wide vs. narrow beach) (Bb bottom plot).

Conceptual Upland and Beach Response Model

Results from the physical analysis of beach width versus upland property show that under both a narrow and wide beach width condition, under a managed retreat scenario, erosion of the upland development could impact up to three parcels inland and in some cases on the inland side of Seacoast Drive. It was assumed that existing armoring structures would be removed in 2030 regaining placement loss (8 m) and initiating managed retreat at an equilibrium width of 25 m (narrow) to 55 m (wide) into the upland development (Figure 12b). In addition, our analysis factored in the costs of removing structures, roads, pipes, and water pumps, as required for managed retreat.

There are many ways to implement a managed retreat using policy such as a buyouts of repetitive loss properties, or acquisition such as a fee simple or outright purchase approach. This project innovated by analyzing the financial viability of public acquisition of vulnerable properties (mostly residential) with a leaseback or long-term rental option so

that the City could recover a portion of the investment before the structures would have to be removed. The leaseback option and some of the considerations are discussed in more detail below.

Economic and Ecological Considerations

Our economic analysis incorporated the costs of structure removal (demolition costs) for all public infrastructure and private property examined in this study, for an estimated USD 8 million. Costs associated with land or easement acquisition and construction costs were not included and would be a future benefit to this type of analysis. The City and adjacent habitats provide a major training site for the U.S. Navy. Costs of removing or relocating federal government property were not considered. Moving military operations elsewhere may be costly, but the loss of the coastal habitats that provide the training grounds could diminish military readiness, and compromise national defense should such a move be required.

Managed retreat preserves more of the recreational value and ecosystem service value than armoring (USD 811 million and USD 56 million by 2100, respectively, for the wide beach scenario. For the narrow scenario, USD 461 million and USD 21 million by 2100), however, the beach is allowed to erode and therefore there is a greater detrimental impact than scenarios involving nourishment cycles. However, the overall losses to private assets for this scenario are approximately USD 176 million, far higher than other alternatives.

Alternative Policy Approach: Buyout with a Leaseback Option

Implementation of managed retreat can take many forms, but one critical issue is how to finance retreat. By its nature, managed retreat involves rezoning public and private property, both land and structures, and eventually removing/modifying many of these structures. Private homeowners in particular have been concerned about the potential loss of property rights, as have business owners with property at risk (e.g., hotels on the coast). One method often discussed [6], but rarely analyzed, is a (fee simple) buyout with a leaseback option. This approach was analyzed using the framework because Imperial Beach, like many coastal cities in California, likely has 30–40 years before damage become a serious concern and the impacts of adaptation strategies on recreation and ecosystem services will be clearly realized.

This approach considers that renters, such as those involved in a leaseback arrangement, may be more amenable to adaptation than homeowners: Renting offers a number of adaptation benefits relative to owning [59]. Renters are more geographically mobile because they face lower migration costs [59]. Once a trigger point is reached, and property must be abandoned, it is far easier for renters to move than homeowners who live on their property. Following the application of this framework analysis, the City may want to consider a short-term rental program, which may generate even higher revenues as well as substantial transient occupancy taxes for the City. A short-term rental program for publicly owned properties at risk may also create less political opposition to retreat because tenants are transient, although short-term rentals may face other political opposition.

Figure 13 provides our estimate of the payback time for fee simple acquisition if the property is leased out to the existing owners—or a new tenant—at current rates. Our analysis assumed that the City could finance a buyback using municipal bonds at a rate of 2.5% a year, which is in line with current market rates for California municipal bonds. Assuming maintenance costs of 5% per year, and given current interest rates, it would take approximately 25 years to pay back a property including both principal and interest, similar to a 30 year home mortgage.

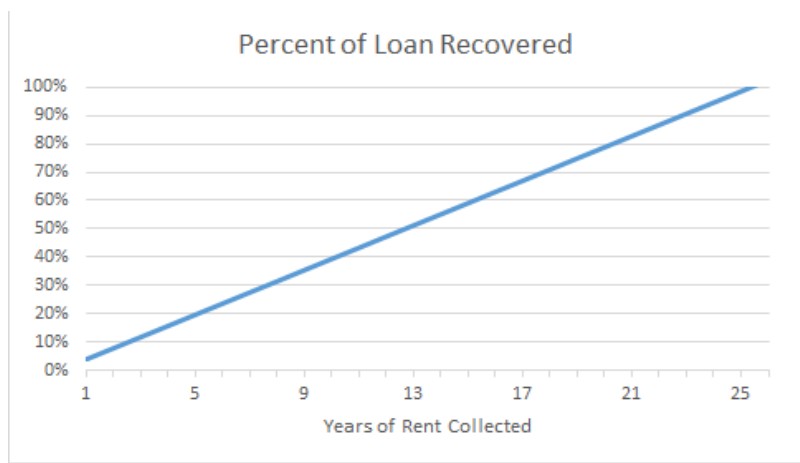

**Figure 13.** Length of time for rent to pay back the property value.

As indicated in Figure 13 above, at the current price/rent ratio (17.5) and current mortgage rates (2.5%), the payback is about 25 or 26 years. Because these parameters may change over time, a simple sensitivity analysis for various combinations of interest rates, maintenance costs, and property tax revenues was conducted (Table 5). The payback time varies depending upon the interest rate used (higher interest rates imply a longer payback), and the ratio of the price of home prices to rents. In the table below, all of the payback times vary between 21 and 45 years, well within the time frame for retreat and on average about the same length as a 30-year mortgage. Higher maintenance fees and including property taxes does not significantly alter payoff time. (Both public property and property owned by nonprofit organizations are not subject to property tax in California.).

Given the time frames involved in adaptation planning, a 25- to 40-year payback period for a public buyout/leaseback option may be feasible given the current forecasts. Of course, the lender, government agency, or nonprofit organization would also likely have to assume the risk that the payback period might be shorter than 30 years. However, if the payback period is longer (i.e., retreat can be staved off a few more years), the property trustee would earn more revenue. It should be emphasized here that for these leaseback programs to be financially viable, local communities must act reasonably soon to acquire properties and develop a long-term or short-term lease/rental market. By 2030, a leaseback program may not be financially viable.

**Table 5.** Payback time (years) of a leaseback arrangement under differing conditions.

| Interest Rate: | | 2.5% | | 4.0% | |
|---|---|---|---|---|---|
| Maintenance Cost: | | 1.0% | 5.0% | 1.0% | 5.0% |
| Prop. Tax: | | 0.0% | 1.2% | 0.0% | 1.2% |
| Price/Rent Ratio | 15 | 21.6 | 22.7 | 26.0 | 27.5 |
| | 17.5 | 25.1 | 26.5 | 30.4 | 32.1 |
| | 20 | 28.7 | 30.3 | 34.7 | 36.6 |
| | 22.5 | 32.3 | 34.1 | 39.1 | 41.2 |
| | 25 | 35.9 | 37.9 | 43.4 | 45.8 |

*3.3. Comparing Adaptation Strategies*

3.3.1. Economic and Fiscal Impacts

The vast majority of taxes generated by beach tourism are TOTs, even though only a small fraction of visitors (less than 25%) stay overnight. Local sales tax revenues are relatively small, reflecting the fact that visitors to Imperial Beach spend relatively little in town, and sales taxes in California do not apply to grocery purchases. As expected, the nourishment strategies, which lead to wider beaches and increased tourism, yield the highest tax revenues. In many ways, these results mirror our other net benefits, with

armoring providing the lowest TOT and sales tax revenues (Figure 14). These tax revenue results are driven largely by spending, which is tied to beach width and recreation.

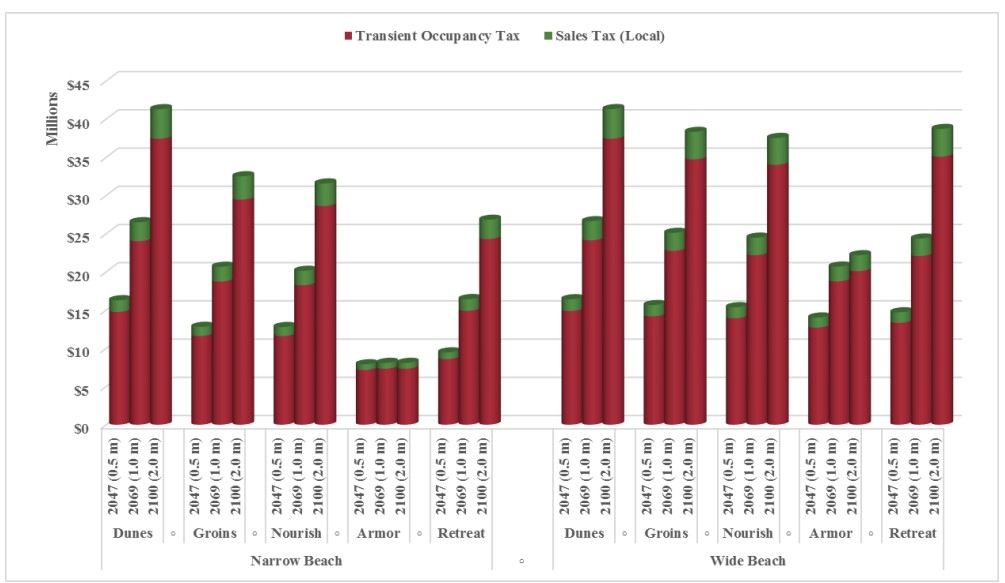

**Figure 14.** Estimated (local) sales tax and transient occupancy tax revenues for different scenarios.

### 3.3.2. Ecology: Replacement Cost Approach

The framework for valuing beach ecosystem services other than recreation using a restoration cost approach is discussed in detail in Section 2.3.2. Figure 15 presents our estimates of the ecological value of Imperial Beach's beaches based on this replacement cost approach as applied to our physical and economic framework. Our results indicate that armoring leads to the lowest value of ecological services for both wide and narrow conditions. Nourishment also provides fewer ecological benefits than dunes, or retreat, or groins, largely due to the recovery time and increasingly frequent nourishment cycles. Indeed, according to our estimates, a policy of dune restoration would provide USD 38.5 million in ecological benefits with 2 m of sea level rise compared to only USD 20.2 million for armoring, a benefit to public trust ecological services of USD 18.3 million.

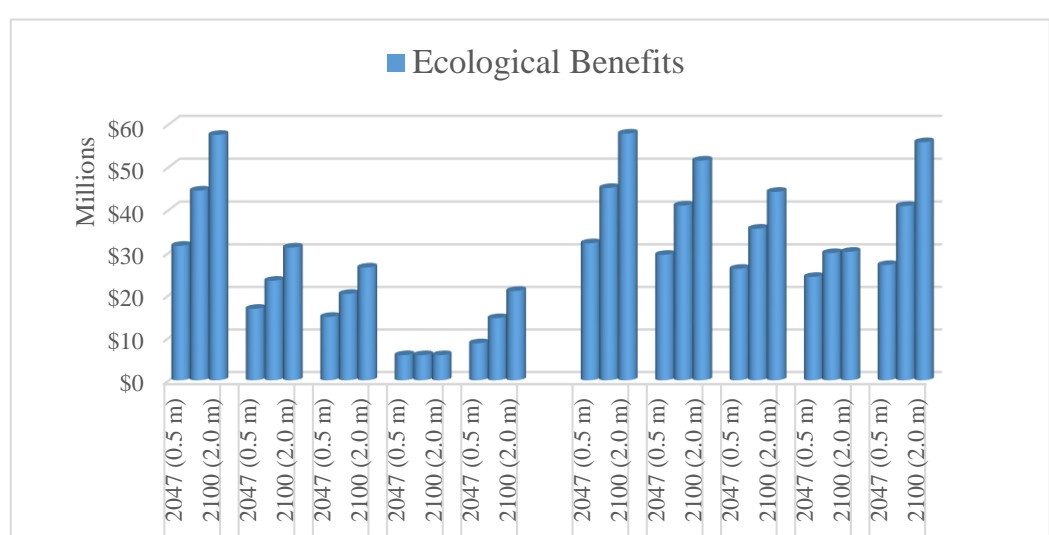

**Figure 15.** Ecological value of beaches in Imperial Beach using the upland vs. beach width framework coupled with a replacement valuation.

Given the uncertainties in our model, these results should be interpreted in general terms. As one would expect, armoring leads to the degradation of ecological services, and this degradation increases with time. Indeed, by 1 m of sea level rise, there are few ecological services left, as the beach has eroded. On the other hand, nourishment, nourishment with groins, dune restoration, and retreat all provide significant ecological services even with 2 m of sea level rise. Beach nourishment and nourishment with groins provide somewhat lower ecological services due to the negative impacts of these policies on beach ecology.

### 3.3.3. Net Benefits

The comparative results of the comprehensive framework evaluation are presented in Figure 16, providing a summary of the net benefits (benefits minus costs) for each adaptation strategy over time. Results differ based on the initial width of the beach assumptions.

For a narrow beach, nourishment strategies dominate and armoring yields much lower net benefits over time because the beach is lost. Retreat provides higher benefits than armoring but lower benefits than nourishment. With a narrow beach, dune restoration yields the highest net benefits in the short-, medium-, and long-term, with armoring yielding the lowest benefits.

For a wide beach over the long-term, net benefits for two strategies, groins and managed retreat, are highest. Although groins yield a slightly higher value, this is well within the margin of error. Nourishment and dunes provide somewhat lower net benefits over time due to increasing frequency of construction and maintenance. In contrast, the net benefits of armoring for both a narrow and wide beach diminishes over time as beach width is eroding, reducing both recreational and ecological value.

To account for the inherent uncertainty in sea level rise projections and analyses of human behavior, a sensitivity analysis was performed on the values assigned to many of our key variables, including the discount rate recreational value, attendance, ecosystem service value (other than recreation or storm buffering) and nourishment/restoration costs. A "robust" result is defined as one in which changes in a particular parameter value do not affect the order of adaptation strategies. Table 6 discusses these sensitivity analysis results for five variables: the discount rate, the (day-use) recreational value, beach attendance, ecosystem service value, and nourishment/restoration costs. These results are discussed in more detail (including diagrams and charts) in the full report [22]. The sensitivity analysis was robust for small changes in the discount rate (between 0% and 5%) However, increasing the (day-use) value of recreation, or if attendance increases, made adaptation strategies using nourishment (e.g., nourishment, living shoreline, groins) a more attractive option in the longer term. Increasing (by 100%) or decreasing (by 50%) the value of ecosystem services did not impact our results. Although our ecological service value estimate and metric are controversial, they did not alter the rankings of adaptation strategies by net benefits. However, this framework approach applying some value for the ecological services not typically included in an assessment of benefits is important for future studies. It is clearly worth more than zero but needs more research and refinement. Increasing the construction costs of nourishment had little impact over longer time horizons.

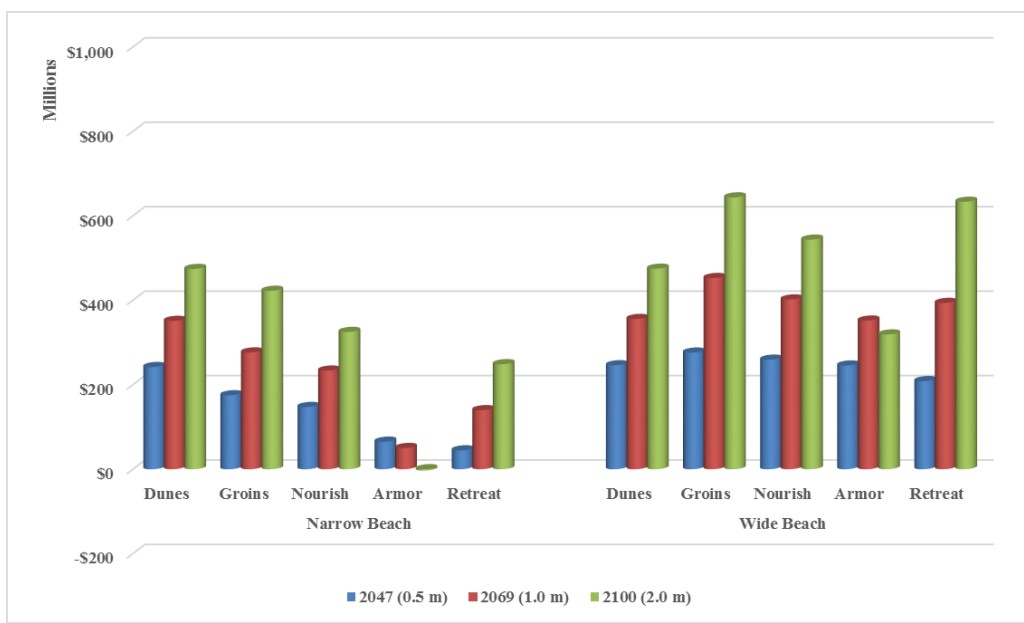

**Figure 16.** Net benefits of the selected adaptation alternatives over time for both a narrow beach (**left**) and a wide beach (**right**).

**Table 6.** Results of Sensitivity Analysis.

| Parameter | Results |
|---|---|
| Discount Rate | Results are robust within a reasonable range (0 to 5%). |
| Recreational Value | Increasing recreational value makes restoration costs (dunes, groins, nourishment) a more attractive option relative to retreat or armoring. |
| Beach Attendance | Increasing beach attendance makes restoration costs (dunes, groins, nourishment) a more attractive option relative to retreat or armoring. |
| Ecosystem Service Value | Results are robust within a reasonable range (0 to 200%) |
| Nourishment/Restoration Costs | Relatively robust over long term |
| Beach Width | Not robust |

In general, results were relatively robust within reasonable changes in the values of the parameters used, with the exception of beach width, which led to significant differences in net benefits with different adaptation strategies, particularly with respect to managed retreat.

In summary, the study's holistic framework analyses indicates that armoring yields the lowest net benefits compared to all other strategies, particularly in later time horizons. As sea level rise increases coastal erosion and other hazards, losses to public trust beach recreation and ecosystem services show that coastal armoring is the poorest long-term adaptation strategy if one considers public trust recreation and ecosystem services.

## 4. Discussion

This innovative adaptation framework is noteworthy for integrating physical changes from different adaptation approaches that affect both upland widths (property and infrastructure damages) with beach widths (public trust recreation and ecological services) over time at a local level. The framework sets up an approach to equitably compare via economic net benefits, the future effects of adaptation choices on all coastal environments and communities attempting to address sea level rise and coastal hazards.

Results show that by including estimates of the value of public trust resources changing over time in response to various adaptation strategies, the most expensive long-term choice is coastal armoring. This result is due to the loss of beaches over time that impacts beach recreation and ecosystems [3,4,18,19,52]. Unfortunately, the last several decades of coastal management decisions have relied on this adaptation strategy to protect upland private development and public infrastructure.

This framework case study application broke ground in two significant ways—a replacement approach to valuing ecosystem services and an economic valuation of a hybrid buyout/leaseback adaptation approach to implementing managed retreat.

First, by applying replacement costs to valuing ecosystem services in the CBA, the approach attempted to include all ecosystem services associated with beaches [1,58]. To our knowledge, this was the first time that a replacement cost approach has been used to estimate the suite of ecosystem services of beaches into adaptation planning. Previous CBAs of sea level rise adaptation only estimates some of the ecological benefits of beaches, and therefore undervalues beaches in comparison to competing uses along the coast (e.g., armoring to protect inland and upland property) [8,10–14]. As explained in more detail in King et al. [47], this approach applies the same valuation metric—replacement cost—used to value human-made structures such as buildings and water pumps, as well as privately owned land.

This approach is controversial. However, sea level rise and other anthropogenic factors imply that the State of California will lose a significant percentage of its beaches during this century [60]. Restoring these beaches and their ecological value will be expensive and applying a lower value other than replacement cost (i.e., the costs of beach restoration) will, in our opinion, lead to an undervaluation of coastal ecology and habitat.

The State of California has prioritized greener, nature-based solutions to protect its public coastal resources against sea level rise [2]. In order for these solutions to "pencil out" under a benefit–cost framework, ecosystem services need to be valued properly and appropriately, which this study endeavors to accomplish using replacement cost [1,18,47] to incorporate the full value of beach ecosystem services into sea-level rise adaptation planning.

With the inclusion of ecosystem services, living shoreline approaches provide significantly higher recreation and ecosystem values over time, particularly in narrower beach condition. With a wider beach condition, managed retreat shows to be the most cost-effective approach over time.

Second, this study also broke ground by economically evaluating a possible implementation mechanism for managed retreat—a public buyout with a leaseback option [16,59]. This strategy was prioritized for detailed analysis by the steering committee primarily to provide an end member of possible ranges of adaptation strategy. Results from this case study suggest adaptation approaches that the City could use to apply both projects and policy changes to address climate change. Managed retreat in particular is challenging because of societal perceptions. Private property owners often feel that they have a right to protect their property, while public members feel that they deserve access to their public trust resources and are often extremely reluctant to pay for typically wealthy oceanfront landowners in a buyout scheme [6,16,61].

However, despite some interesting findings from this adaptation framework approach, there are many improvements that could be made to incorporate recent research advancements and improve the scientific and economic accuracy as well as improve the spatial resolution of the study. These improvements are discussed in the following sections focused on enhancements to the physical modeling and then the economic valuation.

### 4.1. Improvements to the Physical Model

The physical model that modeled the geomorphic evolution of upland and beach widths was driven on the basis of historical erosion rates accelerated by sea level rise. This approach did not allow for inclusion of episodic storm impacts that have been observed to reduce beach widths by 33 m in a single storm event [24,26,27]. In addition, these historical

rates were somewhat influenced by periodic nourishment projects in the littoral cell [26,27]. These rates differ from other published erosion rates [26] for shorter time periods. A higher erosion rate would change the lifecycle costs for all of these projects.

A better approach would be to improve the modeling parameters using some of the recent model development work by USGS [34] or more likely from Scripps Institution of Oceanography around wave run-up [28] and integrating the wave run-up into the storm response of the beach [25]. These models still need refinement to specifically track changes in beach and upland widths through time, as well as to integrate changing elevation surfaces representative of potential adaptation strategies—an exciting opportunity for future collaborations.

### 4.2. Improvements to the Economic Model

As with any study of sea level rise, this study involves assumptions necessary to simulate reality [62]. In particular, assigning monetary value to ecosystem services remains a challenge. Many of the methods used in this analysis are standard (or were standard at the time of the study) [10,12,35,62]. The analysis of flooding, erosion, and the recreational value of beaches is based on relatively standard methods. Our method of estimating the ecological services of beaches beyond recreation and storm buffering is innovative, but also controversial, and the estimates of the offsets used were very conservative. When applied in the United States and elsewhere to mitigate for wetlands loss, an offset is typically several times larger than the loss, implying a *higher* cost.

Assumptions in this study about offsets are much more conservative, in part due to the fact that the framework approach already directly accounts for two key ecosystem services of beaches: recreation and storm buffering. However, for all other ecosystems services of beaches following discussions with City, key local engineers, and ecologists, a good proxy for these beach ecological services was equivalent to *one half of one percent of the cost of the restoration per year*. Again, this could be much higher as is typically used in wetland mitigation.

The purpose of this study was to lay out a framework to consider ecosystem services holistically in adaptation planning and not to defend a specific value; whether the exact dollar value of ecological services is USD 30,000 per acre per year or more or less is subject to debate among researchers, policy makers, and the U.S. Department of Defense. What should not be subject to debate is that beach ecosystem services have value. Historically because it is difficult, ecosystem services have been left out of local adaptation planning and benefit–cost analysis. Future research is needed to refine this valuation method and identify appropriate values for each of the ecosystem services [3]. For example, the length of coastline is also a relevant parameter not included here [47]. Currently, policy makers are willing to consider replacement costs for seawalls and private property structures but not for coastal ecosystems. While controversial, an ecological services value of 0.5% per year of replacement cost seems a reasonable estimate and in the authors' opinion, may very likely be *too low*.

The CSBAT model used to analyze recreational benefits of adaptation alternatives could be improved as well. At the time, it was the standard in California; however, the techniques have evolved since the case study [63]. Further recreational information could expand the detail assigned to each of the widths shown in Figure 5, with more detail on the yard improvements, expenditures by individual recreational users groups of the dry sand beach (e.g., towel space, volleyball, and sandcastle contest), and uses of the intertidal (e.g., surf fishing, beach walking, skim boarding) and subtidal (e.g., surfing, windsurfing, and diving) zones. This improved detail would provide even more site-specific evaluations. However, we do not believe any of these changes would alter our conclusions. The one issue not evaluated that could be significant is the U.S. Department of Defense's role at Imperial Beach, where significant training occurs for the Navy Seals. The loss of training grounds for national defense is extremely difficult to assess and well beyond the scope of this study.

Finally, additional analyses related to physical and social change assumptions and further economic analyses that consider a much broader scientific, engineering, ecological, and social science evaluation are needed. Results from this case study application of the framework illustrates, however, that the long-standing preference for protecting private property at the cost of our public coastal beaches and habitats is the costliest adaptation approach to take in the long term.

*4.3. Applying This Approach to Adaptation Planning*

Local communities in California and elsewhere must make decisions to protect their coastlines. Failure to evaluate all recreation and ecosystem services provided by beaches will lead to undervaluation of these resources with likely long-term detrimental adaptation approaches being implemented.

This framework approach offers an improved method of CBA for adaptation planning at the local level. This holistic method allows public decision makers to make better informed choices about adaptation based on the actual tradeoffs of different strategies and of losing public trust resources [2,19]. The methods employed here to assess the replacement costs of ecosystems is very similar to the methods employed in engineering cost studies—that is to estimate the costs of replacing an asset. The main difference is that we have used a very small percentage of replacement cost as a proxy for ecosystem services other than recreation or storm buffering.

Underestimating the value of coastal ecosystems will lead to under-provision of these services. Although this approach is controversial, we recommend that the State of California and other entities consider the true replacement cost of any ecosystem when evaluating policy alternatives in the future. Coastal ecosystems are under threat not only from climate change, but also from human efforts to forestall its effect [42,64]. This study method and those like it can provide evidence to properly consider long-term impacts to beaches and weigh adaptation options against short-term protection.

In addition to valuing some ecosystem services in a novel way, the study examined a newer policy alternative for implementing managed retreat: a lease buyback program [16]. Crucially, our analysis indicates that given current market prices and interest rates in the City, a lease back arrangement would pay for itself in 20–40 years, which is feasible for many parts of Imperial Beach that are not expected to have to retreat until the middle of the 21st century. According to our sensitivity analysis, even including higher maintenance fees and property taxes for leased property would not significantly alter payoff time. This framework approach to adaptation planning should be considered elsewhere.

## 5. Conclusions

Our analysis indicates the importance of planning ahead. The overall results show that adaptation approaches that prioritize maintaining public beach recreation and ecosystem values in addition to typical considerations of private property losses provide long-term benefits. Hold-the-line armoring to protect private property is not a long-term answer for smaller communities that depend on the beach for their identity and livelihood. For many of these communities, like Imperial Beach, losing the beach means losing revenue. Our examination of changes in TOTs and sales taxes indicates that continued armoring will substantially reduce the City's ability to finance adaptation because it leads to lower sales and TOTs.

These results discourage coastal armoring, indicating implementation of managed retreat substantially increases public revenues over the long-term. We suggest a leaseback program. Implementing a leaseback program today, however, requires that structures would be expected to survive erosion for decades. Waiting to adapt will cost the City valuable time and increase adaptation costs. Local communities must act reasonably soon to acquire properties and develop a long-term or short-term lease/rental market. By 2030, a leaseback program may not be financially viable. To implement this solution, a reasonable approach to incentivizing and converting private to public ownership needs to be identified

so that Imperial Beach and all of our low-lying coastal cities can reduce public liabilities and infrastructure costs, and continue to have a beach to support locals, visitors, and revenues, as well as our flora and fauna into the future.

Building on an in-depth case study in Imperial Beach, this methodology could be extended to aid in the decision-making process for coastal communities worldwide by providing a framework that supports an evidence-backed evaluation of tradeoffs for upland properties versus public trust beach recreation and ecosystem service resources.

**Author Contributions:** A consortium of contributors completed this research. D.R. led the overall project, providing guidance, technical review of all analyses, and writing. P.K. led the economic portions working with J.G., J.C., and A.S., and M.J. provided GIS analysis critical to the project. S.J. provided writing, technical review, and reorganization in responding to comments. J.E. provided engineering construction cost and maintenance estimates important to the cost–benefit analysis. J.N. (retired) and C.H. facilitated critically important City department and community input into the project. Many others supported the work and are identified in the Acknowledgements. All authors have read and agreed to the published version of the manuscript.

**Funding:** This project received funding from the California Coastal Conservancy and the San Diego Foundation.

**Institutional Review Board Statement:** Not applicable.

**Informed Consent Statement:** Not applicable.

**Data Availability Statement:** Data available from City of Imperial Beach, https://www.imperialbeachca.gov/sea_level_rise, accessed on 8 May 2021.

**Acknowledgments:** The authors would like to acknowledge the contributions of City of Imperial Beach City Council Members—Mayor Serge Dedina and Councilman Ed Spriggs; City of Imperial Beach Staff—Andy Hall, City Manager, Hank Levien, Russell Mercer, John French, Robert Stabenow, Steven Dush and Tania Moshirian; regional stakeholders including Port of San Diego—Phil Gibbons and Dennis Larson (NEXUS Planning); Navy—Walt Wilson; SPAWAR—Bart Chadwick; TRNERR—Jeff Crooks and Dani Boudreau; San Diego Regional Climate Collaborative—Laura Engeman; Wildcoast—John Holder; California State Parks—Chris Peregrin; and USC Seagrant—Phylis Grifman and Juliette Finzi-Hart.

**Conflicts of Interest:** The authors declare no conflict of interest. The funders had no role in the design of the study; in the collection, analyses, or interpretation of data; in the writing of the manuscript, or in the decision to publish the results.

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
