# Peer review of "A Holistic Framework for Evaluating Adaptation Approaches to Coastal Hazards and Sea Level Rise: A Case Study from Imperial Beach, California"

_water, doi:10.3390/w13091324_

Round 1

Reviewer 1 Report

See attached comments

Reviewer 2 Report

Paper summary:

In this paper, the authors present a study on adaptation strategies to sea level rise in Imperial Beach, California. Predicting future costs and benefits from coastal adaptation and mitigation strategies is crucial to assist coastal communities in making decisions about their shoreline under SLR. This study uses a physical model and economic analysis to address 5 possible coastal adaptation strategies commonly used along the coast of California. They find that while shore-term hardening strategies are cost effective, ultimately living shorelines and coastal retreat retain the most coastal benefits through maintaining a wider beach.  This paper addresses an important topic in coastal planning and climate change.

General comments:

Overall, I have 5 major comments about the manuscript. First, the paper suffers from a lack of organization. Particularly in the methods, it is hard to follow what is happening. I suggest providing a work flow diagram. Second, the methods do not give enough technical description. For example, there is no description of how the model is run or with what software. Third, the paper needs to be situated within the literature both in the introduction and discussion (add citations and text). The introduction only really has one paragraph of introductory material with 7 paragraphs of site description (which should be in the methods). How does this study fit in with other literature? How does it build upon it? The discussion could also use the same connection to the broader body of literature. Lastly, the conclusion is very repetitive to the discussion - consider how to make the two distinct.

Specific Comments:

Lines 130-137: How did the authors choose which aspects to put into the vulnerability analysis. Did these come from the literature? Why leave out hospitals for example?

Section 2.2.1 This section needs more technical detail about the physical model. For example, how is it implemented – what programs are you using?

Lines 288-308: There seems to be a convolution of beaches and other coastal ecosystems in the coastal habitat model (i.e. carbon sequestration is a mangrove/marsh function)– but yet most of this about beaches (the author’s conceptual model doesn’t have wetlands or other types of ecosystems in it). I am confused by why the authors are using such broad ecosystem functions and basing the valuation on wetlands. Did the authors differentiate between ecosystems? Also what type of ecosystem restoration?

Table 3 – More detail needed - how are the units determined?

Line 482 – I would rename this section. Having a second results section is confusing.

Line 610-612: Why is nourishment incorporated in some of these scenarios (i.e. the groins)? How does that complicate the results of this study?

Figure 16- what do the colors mean? Also red and green is a problematic combination for those with color blindness.

Round 2

Reviewer 2 Report

The authors seem to make many of the changes I asked for. Overall, the methods have a lot more information and the introduction and discussion seem to do a better job of situating the paper in the literature. Unfortunately, the response to comments that was available to me did not include response to my review, nor was there a track changes version of the document – so I was unable to verify the actions of the authors on all of my comments.

Suggested changes:

  • Lines 228-237: what datasets did these variables come from?
  • Table 3 – still can’t find an explanation of how the units of each were determined?
  • There is no Figure 16 anymore? Either replace or renumber.

Round 3

Reviewer 1 Report

The authors responded satisfactorily to the reviewers' comments.